

# Photoemission spectral functions from the three-body Green's function

Gabriele Riva[1,3]⋆, Timothée Audinet[1], Matthieu Vladaj[1],
Pina Romaniello[2,3] and J. Arjan Berger [1,3]†

**1** Laboratoire de Chimie et Physique Quantiques, CNRS, Université de Toulouse,
UPS, 118 route de Narbonne, F-31062 Toulouse, France
**2** Laboratoire de Physique Théorique, CNRS, Université de Toulouse,
UPS, 118 route de Narbonne, F-31062 Toulouse, France
**3** European Theoretical Spectroscopy Facility (www.etsf.eu)

⋆ griva@irsamc.ups-tlse.fr, † arjan.berger@irsamc.ups-tlse.fr

## Abstract

We present an original strategy for the calculation of direct and inverse photo-emission spectra from first principles. The main goal is to go beyond the standard Green's function approaches, such as the *GW* method, in order to find a good description not only of the quasiparticles but also of the satellite structures, which are of particular importance in strongly correlated materials. To this end we use as a key quantity the three-body Green's function, or, more precisely, its hole-hole-electron and electron-electron-hole parts, and we show how the one-body Green's function, and hence the corresponding spectral function, can be retrieved from it. We show that, contrary to the one-body Green's function, information about satellites is already present in the non-interacting three-body Green's function. Therefore, simple approximations to the three-body self-energy, which is defined by the Dyson equation for the three-body Green's function and which contains many-body effects, can still yield accurate spectral functions. In particular, the self-energy can be chosen to be static which could simplify a self-consistent solution of the Dyson equation. We give a proof of principle of our strategy by applying it to the Hubbard dimer, for which the exact self-energy is available.

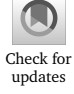

## Contents



# 1   Introduction

Photoemission spectroscopy is one of the most widely used experimental techniques to study the electronic structure of materials [1]. Several photoemission techniques exist and many properties can be obtained like, for example, the band structure of crystalline solids, the binding energies of electrons in molecules such as those involved in chemical bonding, satellites due to (strong) electron correlation etc.. While direct photoemission spectroscopy studies the core and valence states of a material, inverse photoemission spectroscopy studies the unoccupied states. Given the importance of photoemission spectroscopy for the understanding of materials, it is of great importance to complement experiment with theoretical models in order to analyse the experimental data and even to predict these spectra using first-principles methods.

    The most popular first-principles approach to calculate photoemission spectra is many-body perturbation theory based on Green's functions. The main reason is that the one-body Green's function (1-GF) can be easily linked to photoemission spectra (within the sudden approximation) since its poles are the electron removal and addition energies. The 1-GF describes the propagation of a single hole or a single electron in a many-body system. Therefore, all the many-body effects are only implicitly included. In practice the 1-GF is most often obtained from the Dyson equation $G_1 = G_1^0 + G_1^0 \Sigma_1 G_1$, where $G_1$ is the 1-GF, $G_1^0$ is the noninteracting 1-GF and $\Sigma_1$ is the self-energy (1-SE), an effective potential that includes all the many-body effects, which in practice has to be approximated. While there exist approximations to the self-energy that can accurately and efficiently describe quasiparticle energies, e.g., the $GW$ approximation [2], (at least for weakly/moderately correlated systems), the description of satellites, which are a signature of electron correlation in a many-body system, is problematic. In order to obtain non-vanishing satellite structures in the photoemission spectra the self-energy has to be dynamical, i.e., a function of the en-

ergy. Since the non-interacting 1-GF only contains information about quasiparticles a static self-energy can, at most, correct the energy of the quasiparticles but it cannot create additional excitations. It is not straightforward to find good approximations for the dynamical part of the self-energy. Moreover, a dynamical self-energy is inconvenient from a practical point of view because it makes self-consistent calculations very cumbersome. Although fully self-consistent *GW* calculations have been performed on small atoms and molecules [3–10], there are, to the best of our knowledge, no such calculations for solids. Therefore, whenever self-consistency is important, one usually employs partial self-consistent *GW* methods, e.g. quasi-particle self-consistent *GW*, that use a static approximation to the *GW* self-energy [11–15]. As a consequence, there is no self-consistent *GW* approach that can treat both quasiparticles and satellites in solids.

We note that an alternative to solving the Dyson equation is to make an ansatz for the 1-GF, which is the strategy of the cumulant approach [16, 17]. When combined with *GW* this method has been shown to yield accurate quasiparticle energies as well as plasmon satellites [18–26].

In this work we adopt a completely different strategy to capture the physics of both quasi-particles and satellites. In an (inverse) photoemission process a hole (electron) is created and the system will react to this extra particle, by creating electron-hole pairs. Photoemission spectroscopy could therefore be seen as a three-particle process, the electron or hole that is added plus an electron-hole pair. Therefore, we will study here the three-body Green's function (3-GF) as the fundamental quantity from which to obtain the 1-GF and, hence, photoemission spectra. In particular, we will study the electron-hole-hole 3-GF ($G_3^{ehh}$) and the electron-electron-hole 3-GF ($G_3^{eeh}$) which contain all the required information about photoemission and inverse photoemission spectra, respectively. We note that this is a general strategy: the more information the fundamental quantity contains the less information is required in the effective potential, i.e. the self-energy in our case, to describe the relevant many-body effects. Indeed, we will show that already at the level of the non-interacting 3-GF there is information related to satellites. Therefore, a static self-energy (3-SE) is sufficient to obtain both quasiparticles and satellites in the photoemission spectra. We will then demonstrate how one can retrieve the 1-GF and, therefore, the spectral function (which is related to photoemission spectra), from $G_3^{ehh}$ and $G_3^{eeh}$. We illustrate these principles by studying the symmetric Hubbard dimer at 1/4 and 1/2 filling, for which the exact self-energy is known. In particular, we will show that a static approximation to the 3-SE yields excellent results for quasi-particles and satellites at weak correlation and that the results at strong correlation are still very good. Finally, we note that the three-body Green's function has been employed to describe Auger spectra [27], to study satellite structures and the occurrence of the metal-insulator transition [28], and is related to theories that use composite fermion operators, see Ref. [29] for a recent example.

This paper is organized as follows. In section 2 we discuss the theoretical details of the 3-GF and its link to photoemission spectra. We introduce the symmetric Hubbard dimer in section 3 and we show the results we obtained for the spectral functions. Finally, in section 4 we draw our conclusions and we discuss future perspectives.

## 2 Theory

### 2.1 The three-body Green's function

The 3-GF is defined by

$$G_3(1,2,3,1',2',3') = i\langle\Psi_0^N|T[\hat{\psi}_H(1)\hat{\psi}_H(2)\hat{\psi}_H(3)\hat{\psi}_H^\dagger(3')\hat{\psi}_H^\dagger(2')\hat{\psi}_H^\dagger(1')]|\Psi_0^N\rangle, \qquad (1)$$

where $|\Psi_0^N\rangle$ is the ground state of an $N$-particle system, $\hat{\psi}_H, \hat{\psi}_H^\dagger$ are the annihilation and creation operator, respectively, in the Heisenberg representation and $T$ is the time-ordering operator. We use the short-hand notation $(1) = (r_1, s_1, t_1)$ which are the space, spin and time coordinates, respectively. In the following we will express space and spin coordinates as a single variable, namely $x_1 = (r_1, s_1)$. The 3-GF depends on six times or five time differences when the Hamiltonian is time independent, and the total number of permutations of the field operators in Eq. (1) due to the $T$ operator is $6! = 720$. Depending on the order of the field operators (and therefore of the times) the 3-GF yields different information. In general, it describes the propagation of three particles (electrons or holes) and the 3-GF can therefore be split in four components: $G_3^{hhh}$, $G_3^{eee}$, $G_3^{hhe}$ and $G_3^{eeh}$. In order to make this separation explicit one can rewrite the six time-ordered field operators in Eq. (1) as a sum of products of two terms each containing three time-ordered field operators (see also appendix A). Therefore, $6!/(3!\,3!) = 20$ different couples of three time-ordered operators can be formed, one that corresponds to $G_3^{eee}$, one to $G_3^{hhh}$, nine to $G_3^{hhe}$ and nine to $G_3^{eeh}$. As mentioned in the Introduction, in this work we are interested in describing a charged excitation due to an added electron or hole plus an electron-hole pair. Therefore we will focus here on $G_3^{hhe}$ and $G_3^{eeh}$. In order to have a more compact notation, we use here and in the following $G_3^h$ and $G_3^e$, for $G_3^{hhe}$ and $G_3^{eeh}$, respectively, i.e., the presence of the electron-hole pair is implied. It is instructive to write $G_3^{e+h} = G_3^e + G_3^h$ as a function of the five time differences. It is given by

$$
\begin{aligned}
G_3^{e+h}&(x_1, x_2, x_3, x_{1'}, x_{2'}, x_{3'}; \tau_{12}, \tau_{23'}, \tau_{1'2'}, \tau_{2'3}, \tau) \\
&= i \sum_n X_n(x_1, x_2, x_{3'}; \tau_{12}, \tau_{23'}) \tilde{X}_n(x_{1'}, x_{2'}, x_3; \tau_{1'2'}, \tau_{2'3}) \\
&\quad \times exp[i\tau(E_0^N - E_n^{N+1})]\theta(\tau + F(\tau_{12}, \tau_{3'1}, \tau_{1'2'}, \tau_{31'})) \\
&\quad - i \sum_n \tilde{Z}_n(x_{1'}, x_{2'}, x_3; \tau_{1'2'}, \tau_{2'3}) Z_n(x_1, x_2, x_{3'}; \tau_{12}, \tau_{23'}) \\
&\quad \times exp[-i\tau(E_0^N - E_n^{N-1})]\theta(-\tau + F(\tau_{1'2'}, \tau_{31'}, \tau_{12}, \tau_{3'1})), \quad (2)
\end{aligned}
$$

where

$$
\tau = \frac{1}{3}(t_1 + t_2 + t_{3'}) - \frac{1}{3}(t_3 + t_{1'} + t_{2'}) \quad \text{and} \quad \tau_{ij} = t_i - t_j, \tag{3}
$$

$E_n^N$ is the energy of the $n$th excited state of the $N$-particle system and the function $F$ is defined as

$$
F(\tau_{12}, \tau_{3'1}, \tau_{1'2'}, \tau_{31'}) = \sum_{i\neq j\neq k=1,2,3'} \frac{1}{3}(\tau_{ij} - \tau_{ki})\theta(\tau_{jk})\theta(\tau_{ki}) - \sum_{i\neq j\neq k=1',2',3} \frac{1}{3}(\tau_{ij} - \tau_{ki})\theta(\tau_{jk})\theta(\tau_{ij}). \tag{4}
$$

The amplitudes $X_n$ and $Z_n$ are defined as

$$
\begin{aligned}
X_n(x_1, x_2, x_{3'}; \tau_{12}, \tau_{23'}) &= \sum_{i\neq j\neq k=1,2,3'} (-1)^P \theta(\tau_{ij})\theta(\tau_{jk}) \times \exp\Big[\frac{i}{3}(E_0^N(2\tau_{ij} + \tau_{jk}) \\
&\quad + E_n^{N+1}(2\tau_{jk} + \tau_{ij}))\Big]\langle\Psi_0^N|\Upsilon(x_i)e^{-iH\tau_{ij}}\Upsilon(x_j)e^{-iH\tau_{jk}}\Upsilon(x_k)|\Psi_n^{N+1}\rangle, \quad (5)
\end{aligned}
$$

$$
\begin{aligned}
Z_n(x_1, x_2, x_{3'}; \tau_{12}, \tau_{23'}) &= \sum_{i\neq j\neq k=1,2,3'} (-1)^P \theta(\tau_{ij})\theta(\tau_{jk}) \times \exp\Big[\frac{i}{3}(E_0^N(2\tau_{jk} + \tau_{ij}) \\
&\quad + E_n^{N-1}(2\tau_{ij} + \tau_{jk}))\Big]\langle\Psi_n^{N-1}|\Upsilon(x_i)e^{-iH\tau_{ij}}\Upsilon(x_j)e^{-iH\tau_{jk}}\Upsilon(x_k)|\Psi_0^N\rangle, \quad (6)
\end{aligned}
$$

where $P$ is the number of permutations with respect to the initial order $i = 1$, $j = 2$, $k = 3'$. Similarly, the amplitudes $\tilde{X}_n$ and $\tilde{Z}_n$ are defined as

$$
\begin{aligned}
\tilde{X}_n(x_{1'}, x_{2'}, x_3; \tau_{1'2'}, \tau_{2'3}) = \sum_{i \neq j \neq k = 1', 2', 3} (-1)^P \theta(\tau_{ij}) \theta(\tau_{jk}) \times \exp\Bigg[ \frac{i}{3} (E_n^{N+1}(2\tau_{ij} + \tau_{jk}) \\
+ E_0^N(2\tau_{jk} + \tau_{ij})) \Bigg] \langle \Psi_n^{N+1} | \Upsilon(x_i) e^{-iH\tau_{ij}} \Upsilon(x_j) e^{-iH\tau_{jk}} \Upsilon(x_k) | \Psi_0^N \rangle,
\end{aligned}
\tag{7}
$$

$$
\begin{aligned}
\tilde{Z}_n(x_{1'}, x_{2'}, x_3; \tau_{1'2'}, \tau_{2'3}) = \sum_{i \neq j \neq k = 1', 2', 3} (-1)^P \theta(\tau_{ij}) \theta(\tau_{jk}) \times \exp\Bigg[ \frac{i}{3} (E_n^{N-1}(2\tau_{jk} + \tau_{ij}) \\
+ E_0^N(2\tau_{ij} + \tau_{jk})) \Bigg] \langle \Psi_0^N | \Upsilon(x_i) e^{-iH\tau_{ij}} \Upsilon(x_j) e^{-iH\tau_{jk}} \Upsilon(x_k) | \Psi_n^{N-1} \rangle,
\end{aligned}
\tag{8}
$$

where $P$ is the number of permutations with respect to the initial order $i = 1'$, $j = 2'$, $k = 3$. Finally, $\Upsilon(x_i)$ is given by

$$
\Upsilon(x_i) = \begin{cases} \hat{\psi}(x_i) & \text{if} \quad i = 1, 2, 3 \\ \hat{\psi}^\dagger(x_i) & \text{if} \quad i = 1', 2', 3'. \end{cases}
\tag{9}
$$

The details of the derivation of Eq. (2) can be found in Appendix A.

The time $\tau$ in Eq. (3) corresponds to the time of the combined propagation of the added particle (electron or hole) and the electron-hole pair. A Fourier transformation with respect to $\tau$ yields the following expression

$$
\begin{aligned}
& G_3^{e+h}(x_1, x_2, x_3, x_{1'}, x_{2'}, x_{3'}; \tau_{12}, \tau_{23'}, \tau_{1'2'}, \tau_{2'3}, \omega) \\
& = -\sum_n e^{-i[\omega - (E_n^{N+1} - E_0^N)] F(\tau_{12}, \tau_{3'1}, \tau_{1'2'}, \tau_{31'})} \frac{X_n(x_1, x_2, x_{3'}; \tau_{12}, \tau_{23'}) \tilde{X}_n(x_{1'}, x_{2'}, x_3; \tau_{1'2'}, \tau_{2'3})}{\omega - (E_n^{N+1} - E_0^N) + i\eta} \\
& - \sum_n e^{-i[\omega - (E_0^N - E_n^{N-1})] F(\tau_{1'2'}, \tau_{31'}, \tau_{12}, \tau_{3'1})} \frac{\tilde{Z}_n(x_{1'}, x_{2'}, x_3; \tau_{1'2'}, \tau_{2'3}) Z_n(x_1, x_2, x_{3'}; \tau_{12}, \tau_{23'})}{\omega - (E_0^N - E_n^{N-1}) - i\eta}.
\end{aligned}
\tag{10}
$$

From Eq. (10) we see that the first term on the right-hand side corresponds to $G_3^e$ since it has poles at the electron addition energies while the second term on the right-hand side corresponds to $G_3^h$ since its poles are the electron removal energies. The addition (removal) poles are located infinitesimally below (above) the real axis.

The four remaining time differences correspond to the following physical processes: 1) the time between the added particle and the creation of the electron-hole pair ; 2) the time needed to create the electron-hole pair ; 3) the time needed to recombine the electron-hole pair ; 4) the time between the recombination and the removal of the particle. Which time difference corresponds to which process depends on the order of the times. For the description of (inverse) photoemission spectroscopy all four processes can be considered instantaneous. Therefore, we can take the limit $\tau_{ij} \rightarrow 0$ for each of the four time differences. However, the result will depend on the order in which the four limits are taken. It is convenient to choose a time ordering that is coherent with the chronology of the (inverse) photoemission process. For example, in direct photoemission spectroscopy first an electron is emitted from the system leading to the creation of electron-hole pairs. After a time $\tau$ the electron-hole pairs recombine and finally an electron is added. This corresponds to the following order of the field operators $\hat{\psi}^\dagger \hat{\psi}^\dagger \hat{\psi} \hat{\psi}^\dagger \hat{\psi} \hat{\psi}$ that act on $|\Psi_0^N\rangle$. This order of the field operators is obtained with the following choice for the time differences,

$$
\tau_{12} = 0^-, \tau_{23'} = 0^-, \tau_{1'2'} = 0^+, \tau_{2'3} = 0^+.
\tag{11}
$$

From eqs. (5) to (8) one can see that, due to the presence of the Heaviside step functions, only one term in the sum remains after fixing the time differences. We note that

other choices for the time differences are possible to obtain the same order of creation and annihiliation operators mentioned above.

With the time differences given in Eq. (11) we obtain the following expression for $G_3^{e+h}$

$$
\begin{aligned}
&G_3^{e+h}(x_1, x_2, x_3, x_{1'}, x_{2'}, x_{3'}; \omega) \\
&= \sum_n \frac{X_n(x_1, x_2, x_{3'}) X_n^*(x_{1'}, x_{2'}, x_3)}{\omega - (E_n^{N+1} - E_0^N) + i\eta} + \sum_n \frac{Z_n^*(x_{1'}, x_{2'}, x_3) Z_n(x_1, x_2, x_{3'})}{\omega - (E_0^N - E_n^{N-1}) - i\eta},
\end{aligned}
\tag{12}
$$

where the electron-electron-hole and hole-hole-electron amplitudes, $X_n$ and $Z_n$, respectively, are defined as

$$
X_n(x_1, x_2, x_{3'}) = \langle \Psi_0^N | \hat{\psi}^\dagger(x_{3'}) \hat{\psi}(x_2) \hat{\psi}(x_1) | \Psi_n^{N+1} \rangle,
\tag{13}
$$

$$
Z_n(x_1, x_2, x_{3'}) = \langle \Psi_n^{N-1} | \hat{\psi}^\dagger(x_{3'}) \hat{\psi}(x_2) \hat{\psi}(x_1) | \Psi_0^N \rangle.
\tag{14}
$$

For completeness, we also give here the explicit expressions of the complex conjugates of these amplitudes,

$$
X_n^*(x_{1'}, x_{2'}, x_3) = \langle \Psi_n^{N+1} | \hat{\psi}^\dagger(x_{1'}) \hat{\psi}^\dagger(x_{2'}) \hat{\psi}(x_3) | \Psi_0^N \rangle,
\tag{15}
$$

$$
Z_n^*(x_{1'}, x_{2'}, x_3) = \langle \Psi_0^N | \hat{\psi}^\dagger(x_{1'}) \hat{\psi}^\dagger(x_{2'}) \hat{\psi}(x_3) | \Psi_n^{N-1} \rangle.
\tag{16}
$$

The representation of $G_3^{e+h}$ in Eq. (12) is similar to the Lehmann representation of $G_1$, i.e., the poles are the same but the amplitude corresponding to each pole is different. In section 2.2 we will use this similarity to obtain a relation that links $G_3^{e+h}$ to $G_1$.

While the time ordering in Eq. (11) yields the chronological order of the field operators for the electron removal process it does not yield an equivalent order for the electron addition process, which would be $\hat{\psi}\hat{\psi}^\dagger\hat{\psi}\hat{\psi}^\dagger\hat{\psi}\hat{\psi}^\dagger$ acting on $|\Psi_0^N\rangle$, i.e., the creation of an electron that leads to the formation of electron-hole pairs followed by recombination and electron removal. However, our goal is to calculate the spectral function, which requires the knowledge of the 1-GF only. In this case the order of the creation of the particle and the creation of the electron-hole pair is not important. As we will show in the next subsection the exact 1-GF can be recuperated from $G_3^{e+h}$ with the time ordering given in Eq. (11). We note that, alternatively, one could choose two different time orderings for the removal and addition processes. The difference between the two approaches is only that with one time ordering we can write a single Dyson-like equation for $G_3^{e+h}$ while with two time orderings we would need two Dyson-like equations, one for $G_3^h$ and $G_3^e$ each.

## 2.2  Obtaining $G_1$ from $G_3^{e+h}$

As mentioned in the previous subsection, our goal is to calculate the spectral function which is defined in terms of $G_1$. We therefore require an equation that yields $G_1$ from $G_3^{e+h}$. As explained in Appendix B such a relation can be obtained by contracting the position-spin variables of the field operators that correspond to electron-hole pairs followed by integration over the contracted variables, i.e.,

$$
G_1^e(x_1, x_{1'}, \omega) = \frac{1}{N^2} \iint dx_2 dx_3 \, G_3^e(x_1, x_2, x_3, x_{1'}, x_3, x_2, \omega),
$$

$$
G_1^h(x_1, x_{1'}, \omega) = \frac{1}{(N-1)^2} \iint dx_2 dx_3 \, G_3^h(x_1, x_2, x_3, x_{1'}, x_3, x_2, \omega),
\tag{17}
$$

where the integrals include a summation over the spin and $G_1^e$ ($G_1^h$) refers to the addition (removal) part of $G_1$.

## 2.3 Dyson equation

As for the 1-GF, the definition of the 3-GF in Eq. (1) is not useful for practical calculations since its evaluation requires the knowledge of the $N$-body ground state wave function. Similarly, the expression of $G_3^{e+h}$ in Eq. (12) involves the $N$-body ground state wave function as well as excited-state wave functions of the corresponding $N+1$ and $N-1$ electron systems. It is therefore convenient to introduce an effective potential that links $G_3^{e+h}$ to $G_{03}^{e+h}$, i.e., the noninteracting $G_3^{e+h}$. Therefore, in the same spirit as for the 1-GF, we introduce a self-energy $\Sigma_3$ that is defined by the following Dyson equation

$$
\begin{aligned}
G_3^{e+h}(x_1, x_2, x_3, x_{1'}, x_{2'}, x_{3'}, \omega) =& G_{03}^{e+h}(x_1, x_2, x_3, x_{1'}, x_{2'}, x_{3'}, \omega) \\
& + G_{03}^{e+h}(x_1, x_2, x_6, x_{4'}, x_{5'}, x_{3'}, \omega) \\
& \times \Sigma_3(x_{4'}, x_{5'}, x_{6'}, x_4, x_5, x_6, \omega) \\
& \times G_3^{e+h}(x_4, x_5, x_3, x_{1'}, x_{2'}, x_{6'}, \omega).
\end{aligned}
\tag{18}
$$

Here, and in the rest of the paper, integration over repeated indices are implied. As mentioned before, one could also define two self-energies and, hence, have two Dyson equations, one for $G_3^e$ and one for $G_3^h$. Indeed, in the case of the 1-GF it was found that the separate calculation of its addition and removal parts can in some cases lead to improved results [30–32]. Here we focus on a unified description of removal and addition processes since it has the advantage of yielding the full spectral function from a single calculation. It is also the most common approach for the calculation of the 1-GF. We note that we can invert the Dyson equation in Eq. (18) to obtain

$$
\begin{aligned}
[G_3^{e+h}]^{-1}(x_1, x_2, x_3, x_{1'}, x_{2'}, x_{3'}, \omega) =& [G_{03}^{e+h}]^{-1}(x_1, x_2, x_3, x_{1'}, x_{2'}, x_{3'}, \omega) \\
& - \Sigma_3(x_1, x_2, x_3, x_{1'}, x_{2'}, x_{3'}, \omega).
\end{aligned}
\tag{19}
$$

We refer the reader to appendix C for the details.

Any non-interacting $n$-body Green's function can be written in terms of non-interacting 1-GFs for which an analytic expression is well known. In particular, the full noninteracting 3-GF can be written according to [33]

$$
\begin{aligned}
G_{03}(1, 2, 3, 1', 2', 3') =& G_{01}(1, 1')G_{01}(2, 2')G_{01}(3, 3') + G_{01}(1, 2')G_{01}(2, 3')G_{01}(3, 1') \\
& + G_{01}(1, 3')G_{01}(2, 1')G_{01}(3, 2') - G_{01}(1, 1')G_{01}(2, 3')G_{01}(3, 2') \\
& - G_{01}(1, 2')G_{01}(2, 1')G_{01}(3, 3') - G_{01}(1, 3')G_{01}(2, 2')G_{01}(3, 1'),
\end{aligned}
\tag{20}
$$

where $G_{01}$ is the non-interacting 1-GF. Taking into account the choice for the time differences in Eq. (11) and performing a Fourier transform with respect to $\tau$ given in Eq. (3) we obtain the following expression for $G_{03}^{e+h}$

$$
\begin{aligned}
G_{03}^{e+h}&(x_1, x_2, x_3, x_{1'}, x_{2'}, x_{3'}; \omega) \\
=& \int \frac{d\omega' d\omega''}{(2\pi)^2} G_{01}(x_1, x_{1'}; \omega + \omega' - \omega'')G_{01}(x_2, x_{2'}; \omega'')G_{01}(x_3, x_{3'}; \omega') \\
& + G_{01}(x_1, x_{2'}; \omega)G_{01}(x_2, x_{3'})G_{01}(x_3, x_{1'}) + G_{01}(x_1, x_{3'})G_{01}(x_2, x_{1'}; \omega)G_{01}(x_3, x_{2'}) \\
& - G_{01}(x_1, x_{1'}; \omega)G_{01}(x_2, x_{3'})G_{01}(x_3, x_{2'}) - G_{01}(x_1, x_{3'})G_{01}(x_2, x_{2'}; \omega)G_{01}(x_3, x_{1'}) \\
& - \int \frac{d\omega' d\omega''}{(2\pi)^2} G_{01}(x_1, x_{2'}; \omega + \omega' - \omega'')G_{01}(x_2, x_{1'}; \omega'')G_{01}(x_3, x_{3'}; \omega'),
\end{aligned}
\tag{21}
$$

where $G_{01}$ is defined as [34, 35]

$$G_{01}(x_1, x_{1'}; \omega) = \sum_n \frac{\phi_n(x_1)\phi_n^*(x_{1'})}{\omega - \epsilon_n + i\eta \text{sign}(\epsilon_n - \mu)}, \tag{22}$$

$$G_{01}(x_1, x_{1'}) = G_{01}(x_1, x_{1'}, \tau \to 0^-) = i\gamma(x_1, x_{1'}) = i\sum_v \phi_v(x_1)\phi_v^*(x_{1'}), \tag{23}$$

in which $\mu$ is the chemical potential, $\phi_n$ and $\epsilon_n$ are single-particle wave functions and energies, respectively, $v$ corresponds to valence states and $\gamma$ is the one-body reduced density matrix.

In Eq. (21) one can recognize two types of contributions on the right-hand side. The first type contains a product of three noninteracting 1-GFs of which only one depends on the frequency. From Eqs. (22) and (23) we then observe that these contributions correspond to quasi-particles since their poles correspond to a single eigenenergy. The second type contains two convolutions. Let us work out one of these contributions. We obtain

$$\int \frac{d\omega' d\omega''}{(2\pi)^2} G_{01}(x_1, x_{1'}; \omega + \omega' - \omega'') G_{01}(x_2, x_{2'}; \omega'') G_{01}(x_3, x_{3'}; \omega')$$

$$= \sum_v \sum_{c,c'} \frac{\phi_c(x_1)\phi_c^*(x_{1'})\phi_{c'}(x_2)\phi_{c'}^*(x_{2'})\phi_v(x_3)\phi_v^*(x_{3'})}{\omega - \epsilon_c - (\epsilon_{c'} - \epsilon_v) + i\eta}$$

$$+ \sum_{v,v'} \sum_c \frac{\phi_v(x_1)\phi_v^*(x_{1'})\phi_{v'}(x_2)\phi_{v'}^*(x_{2'})\phi_c(x_3)\phi_c^*(x_{3'})}{\omega - \epsilon_v + (\epsilon_c - \epsilon_{v'}) - i\eta}, \tag{24}$$

where $v(c)$ corresponds to valence (conduction) states. From the above expression we see that the poles of this contribution correspond to the sum of an eigenenergy and an eigenenergy difference of a conduction and valence state. This shows that $G_3^{e+h}$ already contains information about satellites in the non-interacting limit. Therefore, even with only a static 3-SE the resulting 1-GF (obtained from Eq. (17)) will, in general, include satellites. The main task of a static 3-SE is to modify the position (and spectral weight) of the poles, both due to quasiparticles and satellites, and bring them closer to the exact removal and addition energies. For these reasons we will focus in the following on a static 3-SE. The Dyson equation thus becomes

$$[G_{3static}^{e+h}]^{-1}(\omega) = [G_{03}^{e+h}]^{-1}(\omega) - \Sigma_3(\omega = 0), \tag{25}$$

where we omitted the spin-position dependence for notational convenience.

## 2.4 The 3-body spectral function

Since the spectral representation of $G_3^{e+h}(\omega)$ given in Eq. (12) is similar to the one of $G_1$ it is convenient to introduce a 3-body spectral function for $G_3^{e+h}(\omega)$ that is similar to the spectral function corresponding to $G_1$. The latter is defined as

$$A(x_1, x_{1'}; \omega) = \frac{1}{\pi} \text{sign}(\mu - \omega) \text{Im} G_1(x_1, x_{1'}; \omega). \tag{26}$$

We can thus define the spectral function $A_3(\omega)$ corresponding to $G_3^{e+h}(\omega)$ according to

$$A_3(\omega) = \frac{1}{\pi} \text{sign}(\mu - \omega) \text{Im} G_3^{e+h}(\omega), \tag{27}$$

where, for notational convenience, the spin-position arguments are omitted. It can be verified that $G_3^{e+h}(\omega)$ can be retrieved from $A_3(\omega)$ according to

$$G_3^{e+h}(\omega) = \int_{-\infty}^{\mu} d\omega' \frac{A_3(\omega')}{\omega - \omega' - i\eta} + \int_{\mu}^{+\infty} d\omega' \frac{A_3(\omega')}{\omega - \omega' + i\eta}. \tag{28}$$

By comparing the above expression to Eq. (12) we see that $A_3(\omega)$ can be written as

$$A_3(x_1, x_2, x_3, x_{1'}, x_{2'}, x_{3'}; \omega) = \sum_n X_n(x_1, x_2, x_{3'}) X_n^*(x_{1'}, x_{2'}, x_3) \delta(\omega - (E_n^{N+1} - E_0^N))$$
$$+ \sum_n Z_n(x_1, x_2, x_{3'}) Z_n^*(x_{1'}, x_{2'}, x_3) \delta(\omega - (E_0^N - E_n^{N-1})). \tag{29}$$

It is easy to show that $A_3(\omega)$, as is the spectral function corresponding to $G_1$, is a hermitian and positive define matrix.

We note that the 3-body spectral function is not the spectral function that corresponds to photoemission spectroscopy. Both spectral functions have the same poles but the corresponding amplitudes are different. In particular, in the non-interacting case the amplitudes related to satellites can be non-zero in the 3-body spectral function. To retrieve the spectral function that corresponds to photoemission spectra Eq. (17) has to be used.

## 2.5 the 3-GF in a general basis

For practical applications it is convenient to express $G_3^{e+h}$ in a basis set. We can write the field operator in a general one-electron basis set $\{\phi_i\}$ according to

$$\hat{\psi}(x) = \sum_i \hat{c}_i \phi_i(x), \qquad \hat{\psi}^\dagger(x) = \sum_i \hat{c}_i^\dagger \phi_i^*(x), \tag{30}$$

where $\hat{c}_i$ and $\hat{c}_i^\dagger$ are the annihilation and creation operator respectively, with the usual anti-commutation relations.

It is then straightforward to show that Eq. (12) can be rewritten as

$$G_3^{e+h}(x_1, x_2, x_3, x_{1'}, x_{2'}, x_{3'}; \omega) = \sum_{ijlmok} G_{3(ijl;mok)}^{e+h}(\omega) \phi_i(x_1) \phi_j(x_2) \phi_l^*(x_{3'})$$
$$\times \phi_m^*(x_{1'}) \phi_o^*(x_{2'}) \phi_k(x_3), \tag{31}$$

where $G_{3(ijl;mok)}^{e+h}(\omega)$ is $G_3^{e+h}(\omega)$ expressed in the basis $\{\phi_i\}$ according to

$$G_{3(ijl;mok)}^{e+h}(\omega) = \sum_n \frac{X_n^{ijl} X_n^{\dagger mok}}{\omega - (E_n^{N+1} - E_0^N) + i\eta} + \sum_n \frac{Z_n^{ijl} Z_n^{\dagger mok}}{\omega - (E_0^N - E_n^{N-1}) - i\eta}, \tag{32}$$

in which

$$X_n^{ijl} = \langle \Psi_0^N | \hat{c}_l^\dagger \hat{c}_j \hat{c}_i | \Psi_n^{N+1} \rangle, \qquad X_n^{\dagger mok} = \langle \Psi_n^{N+1} | c_m^\dagger \hat{c}_o^\dagger \hat{c}_k | \Psi_0^N \rangle,$$
$$Z_n^{ijl} = \langle \Psi_n^{N-1} | \hat{c}_l^\dagger \hat{c}_j \hat{c}_i | \Psi_0^N \rangle, \qquad Z_n^{\dagger mok} = \langle \Psi_0^N | c_m^\dagger \hat{c}_o^\dagger \hat{c}_k | \Psi_n^{N-1} \rangle. \tag{33}$$

Finally, we note that when expressed in a basis set Eq. (17) becomes

$$G_{1(im)}^e(\omega) = \frac{1}{N^2} \sum_{jk} G_{3(ijj;mkk)}^e(\omega), \tag{34}$$

$$G_{1(im)}^h(\omega) = \frac{1}{(N-1)^2} \sum_{jk} G_{3(ijj;mkk)}^h(\omega). \tag{35}$$

# 3  Symmetric Hubbard dimer

In order to illustrate the strategy discussed in the previous section we consider the symmetric Hubbard dimer. It consists of two degenerate sites each containing one orbital; moreover only electrons on the same site interact with each other. This model is exactly solvable and, therefore, allows us to test the accuracy of various approximations in both the weakly and strongly correlated regimes. In particular the exact 3-body self-energy, which is the key quantity to calculate the 3-GF, is available.

We will study the symmetric Hubbard dimer at 1/4 and 1/2 filling.

## 3.1  The Hamiltonian

The Hamiltonian corresponding to the symmetric Hubbard dimer is given by

$$H = -t \sum_{i \neq j=1,2} \sum_{\sigma} c_{i\sigma}^{\dagger} \hat{c}_{j\sigma} + \frac{U}{2} \sum_{i=1,2} \sum_{\sigma\sigma'} c_{i\sigma}^{\dagger} c_{i\sigma'}^{\dagger} \hat{c}_{i\sigma'} \hat{c}_{i\sigma} + \epsilon_0 \sum_{i=1,2} \sum_{\sigma} n_{i\sigma}, \qquad (36)$$

in which $-t, U$ and $\epsilon_0$ represent the hopping kinetic energy, the (spin-independent) on-site interaction and the orbital energy, respectively, and $n_{i\sigma} = c_{i\sigma}^{\dagger} \hat{c}_{i\sigma}$ is the number operator. We made explicit the spin $\sigma$ in the above equation. We note that the amount of electron correlation in the system is proportional to the ratio $U/t$. The eigenenergies and eigenfunctions of the above Hamiltonian can be found analytically. In appendix D we report the eigenvalues and eigenfunctions for the cases of one, two and three electrons. The details of the calculations can be found in, e.g., Refs. [36, 37].

Without loss of generality, we can make use of the following simplifications when evaluating $G_3^{e+h}$ for the Hubbard dimer: 1) the electron involved in the neutral excitation does not change its spin, and 2) the electron or hole that is added to the system is different from the one involved in the neutral excitation.

## 3.2  1/4 filling

In the study of the one-electron case, we focus our attention only on the addition part of the spectral function because its removal part is straightforward since there is no correlation and, therefore, no satellites. Moreover, the removal part of $G_3^{e+h}$, as defined in Eq. (12), vanishes. This is due to the fact that, when the only electron present in the system is removed, there are no electrons left to create the neutral excitations.

In Appendix E we show how the electron-electron-hole Green's function in the diagonal basis is given by the following expression

$$G_3^e(\omega) = \text{diag}(G_3'(\omega), G_3'(\omega), G_3''(\omega), G_3'''(\omega), G_3'^v(\omega)), \qquad (37)$$

where

$$G_3'(\omega) = \frac{1}{\omega - (\epsilon_0 + t) + i\eta},$$

$$G_3''(\omega) = \frac{1}{\omega - (\epsilon_0 + U + t) + i\eta},$$

$$G_3'''(\omega) = \frac{1}{\omega - (\epsilon_0 + \frac{U+c}{2} + t) + i\eta},$$

$$G_3'^v(\omega) = \frac{1}{\omega - (\epsilon_0 + \frac{U-c}{2} + t) + i\eta}, \qquad (38)$$

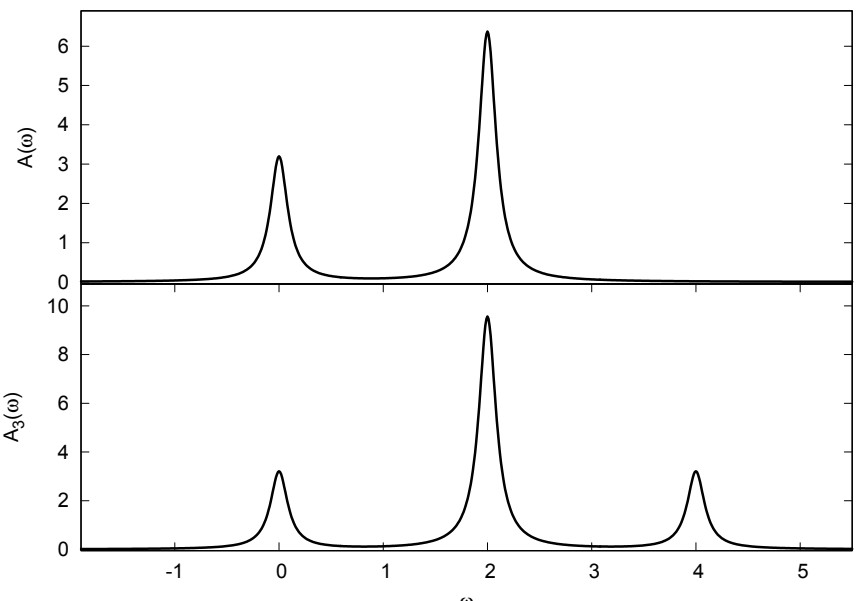

Figure 1: The addition part of the 1- and 3-body spectral function for the Hubbard dimer at 1/4 filling in the non-interacting limit ($U = 0$). Top panel: the exact spectral function $A(\omega)$. Bottom panel: the exact three-body spectral function $A_3(\omega)$. The peak at $\omega = 4$, which is present only in the three-body spectral function, is related to a satellite. The spectra correspond to $\epsilon_0 = 1$.

with $c = \sqrt{16t^2 + U^2}$. We observe that $G_3^e$ contains four distinct poles for $U > 0$. We emphasize that the on-site interaction $U$ only influences the position of the poles but not the corresponding amplitudes. Thanks to this feature, amplitudes related to satellites do not vanish in the non-interacting limit. The non-interacting $G_3^e$ hence reads

$$G_{03}^e(\omega) = \text{diag}(G_{03}'(\omega), G_{03}'(\omega), G_{03}''(\omega), G_{03}'''(\omega), G_{03}'^v(\omega)), \tag{39}$$

with

$$G_{03}'(\omega) = G_{03}''(\omega) = \frac{1}{\omega - (\epsilon_0 + t) + i\eta},$$
$$G_{03}'''(\omega) = \frac{1}{\omega - (\epsilon_0 + 3t) + i\eta},$$
$$G_{03}'^v(\omega) = \frac{1}{\omega - (\epsilon_0 - t) + i\eta}. \tag{40}$$

We see that at $U = 0$ two the poles present at $U > 0$ merge and only three distinct poles remain. Since the energy levels for the Hubbard dimer at 1/4 filling are equal to $\epsilon_0 - t$ and $\epsilon_0 + t$ (see Appendix D) we can therefore conclude that $G_{03}'''$ is related to a satellite since its pole is equal to the sum of $\epsilon_0 + t$, i.e., the energy of the antibonding level, and $2t$, i.e., the energy of a neutral excitation.

Since both $G_3^e$ and $G_{03}^e$ are diagonal also the 3-SE is diagonal, as can be seen from Eq. (19). Moreover, each diagonal element of both $G_3^e$ and $G_{03}^e$ contains a single pole. Therefore the exact 3-SE is static (as it should since one cannot have more than three particles in the system, i.e. the added electron and the electron-hole pair which it creates) and has the following simple expression

$$\Sigma_3 = \text{diag}\left(0; 0; U; \frac{U+c}{2} - 2t; \frac{U-c}{2} + 2t\right). \tag{41}$$

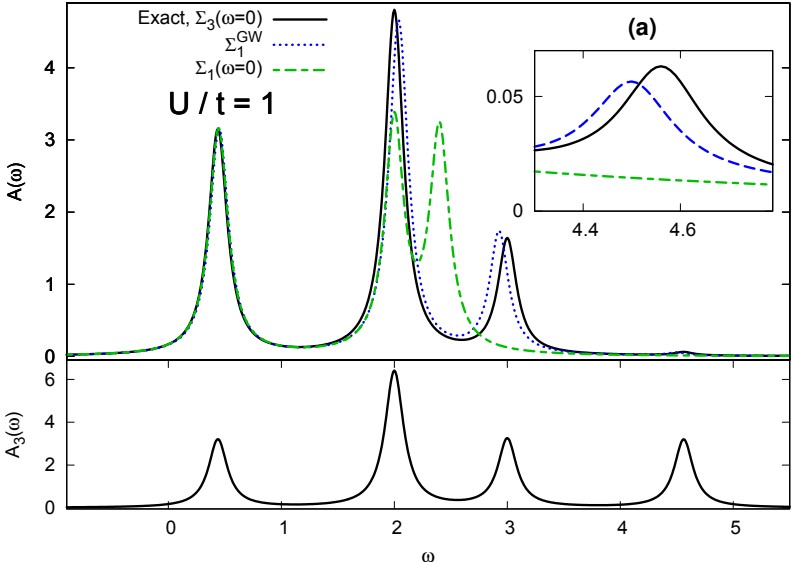

Figure 2: The addition part of the the 1- and 3-body spectral functions for the Hubbard dimer at 1/4 filling at weak interaction ($U/t = 1$). Top panel: the spectral function $A(\omega)$ obtained with various levels of theory: the exact 1-GF and the 1-GF obtained from the exact static 3-SE (black solid line); the *GW* approximation (blue dotted line); the exact static 1-GF (green dashed line). Inset (a): zoom of the satellite peak. Bottom panel: the exact 3-body spectral function $A_3(\omega)$. All spectra correspond to $\epsilon_0 = 1$.

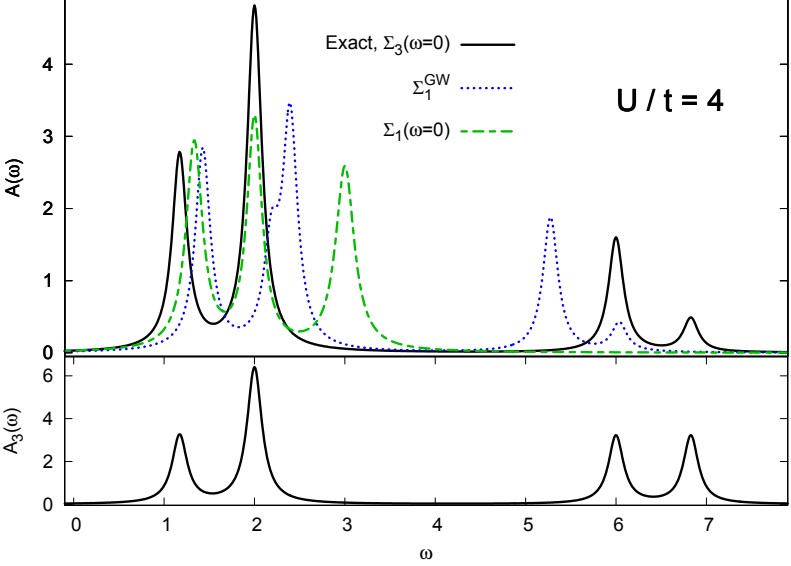

Figure 3: The addition part of the the 1- and 3-body spectral functions for the Hubbard dimer at 1/4 filling at strong interaction ($U/t = 4$). Top panel: the spectral function $A(\omega)$ obtained with various levels of theory: the exact 1-GF and the 1-GF obtained from the exact static 3-SE (black solid line); the *GW* approximation (blue dotted line); the exact static 1-GF (green dashed line). The rightmost peak is a satellite. Bottom panel: the exact 3-body spectral function $A_3(\omega)$. All spectra correspond to $\epsilon_0 = 1$.

To illustrate the difference with standard approaches using the 1-GF, we also report the addition part of $G_1$ and the corresponding 1-SE. In the bonding/anti-bonding basis $G_1^e$ is diagonal and it is given by

$$G_1^e(\omega) = \text{diag}(G_1'(\omega), G_1''(\omega), G_1'''(\omega)), \tag{42}$$

where

$$
\begin{aligned}
G_1'(\omega) &= \frac{1}{\omega - (\epsilon_0 + t) + i\eta}, \\
G_1''(\omega) &= \frac{1}{2}\left( \frac{1}{\omega - (\epsilon_0 + t) + i\eta} + \frac{1}{\omega - (\epsilon_0 + U + t) + i\eta} \right), \\
G_1'''(\omega) &= \frac{\frac{1}{a^2}(A-1)^2}{\omega - (\epsilon_0 + \frac{U-c}{2} + t) + i\eta} + \frac{\frac{1}{b^2}(B-1)^2}{\omega - (\epsilon_0 + \frac{U+c}{2} + t) + i\eta},
\end{aligned}
\tag{43}
$$

with $a = \frac{\sqrt{2}}{c-U}\sqrt{16t^2 + (c-U)^2}$, $A = \frac{4t}{U-c}$, $b = \frac{\sqrt{2}}{c+U}\sqrt{16t^2 + (c+U)^2}$ and $B = \frac{4t}{U+c}$. We observe that $G_1^e$ has four poles, i.e., the same amount as $G_3^e$ as it should. As a consequence, and in contrast to $G_3^e$, some of the terms on the diagonal of $G_1^e$ contain more than one pole. When the interaction is switched off ($U = 0$), the diagonal components of the 1-GF become

$$
\begin{aligned}
G_0'(\omega) = G_0''(\omega) &= \frac{1}{\omega - (\epsilon_0 + t) + i\eta}, \\
G_0'''(\omega) &= \frac{1}{\omega - (\epsilon_0 - t) + i\eta}.
\end{aligned}
\tag{44}
$$

Thus only two distinct poles remain both corresponding to quasi-particles. It can be verified that at $U = 0$ the pole in the second term on the right-hand side of Eq. (43) is equal to $\epsilon_0 + 3t$ which corresponds to the position of the satellite. However, at $U = 0$ the corresponding spectral weight vanishes since $B - 1 = 0$. Therefore, there is no trace of this satellite in $G_{01}$. From the Dyson equation $\Sigma = G_0^{-1} - G_1^{-1}$ it is easy to verify that the self-energy is frequency dependent [36]. We conclude that the exact 1-SE is a more complicated expression than the exact 3-SE.

In Fig. 1 we show a comparison between the spectral function corresponding to the non-interacting one-body Green's functions $G_{01}^e$ and the spectral function corresponding to the non-interacting three-body Green's function $G_{03}^e$, defined in Eq. (27). One can see that, in the non-interacting limit, the peak at the satellite position is present with a nonvanishing spectral weight only in the spectral function obtained from the $G_{03}^e$. This was to be expected from the discussion above. The non-interacting 1-GF is retrieved from $G_{03}^e$ by using Eq. (34).

In Fig. 2 we compare the spectral functions for $U/t = 1$ obtained with $\Sigma_3(\omega = 0)$ and $\Sigma_1(\omega = 0)$, i.e., the exact static approximations to the 3-SE and the 1-SE, respectively. In the former case we used Eq. (34) to retrieve $G_1^e$ from $G_3^e$. For completeness, we also report the corresponding 3-body spectral function in the bottom panel of Fig. 2. At 1/4 filling the spectral function obtained from $\Sigma_3(\omega = 0)$ is exact. Instead, the spectral function corresponding to $\Sigma_1(\omega = 0)$ misses the (small) satellite peak, as was expected, and greatly underestimates the position of the highest-energy quasiparticle peak. We also report the spectral function obtained from a dynamical 1-SE, namely the popular $GW$ approximation to the 1-SE. The analytical result for the $GW$ approximation can be found in Ref. [36]. We see that $\Sigma_1^{GW}$ yields a very good spectral function at 1/4 filling and weak interaction.

When we increase the interaction strength to $U/t = 4$ (see Fig. 3) we observe that the spectral weight of the satellite in the spectral function has also increased. Instead, in the 3-body spectral function the spectral weight related to the satellite is not influenced by the interaction strength, only its position depends on it. Again, after application of

Eq. (34) we retrieve $G_1^e$ from $G_3^e$ which, as mentioned before, leads to the exact spectral function in the case of 1/4 filling. From the spectral function obtained from $\Sigma_1(\omega = 0)$ we observe that the underestimation of the position of the highest-energy quasiparticle peak is even larger than was the case at weak interaction strength. Moreover, it overestimates the position of the lowest-lying quasiparticle peak. Finally, at strong correlation the energies of the quasiparticles and the satellite in the $GW$ spectral functions are either substantially overestimated or underestimated. Moreover, the quasiparticle peak just above $\omega = 2$ is split into two peaks due to a spurious pole in the $GW$ Green's function.

From the above we conclude that the 1-SE has a more difficult task than the 3-SE, since it has to create a satellite which is not present in the $G_{01}$, whereas it is already present in $G_{03}$. This is an important point, because we can hope that simple approximations to $\Sigma_3$ can still produce accurate spectral functions. For example, in the Hubbard dimer at 1/4 filling, the static approximation (25) is exact for the 3-GF. Instead, with a static 1-SE, it is not possible to obtain a nonvanishing satellite amplitude.

## 3.3  1/2 filling

We now study the $G_3^{e+h}$ for the Hubbard dimer at 1/2 filling. We start by considering the process of electron addition and removal separately since in this way both $G_3^h$ and $G_3^e$ can be written in a simple diagonal form. Later we will write the diagonal expression of the total $G_3^{e+h}$. The details of the calculations are given in appendix F.

In the diagonal basis, we obtain the following expressions for the removal and addition parts,

$$G_3^h(\omega) = \mathrm{diag}(0,1,0,1)\frac{1}{\omega - (\epsilon_0 + t + \frac{U-c}{2}) - i\eta} + \mathrm{diag}(1,0,1,0)\frac{1}{\omega - (\epsilon_0 - t + \frac{U-c}{2}) - i\eta}, \tag{45}$$

$$G_3^e(\omega) = \mathrm{diag}(0,\lambda_1,0,\lambda_1)\frac{1}{\omega - (\epsilon_0 + t + \frac{c+U}{2}) + i\eta} + \mathrm{diag}(\lambda_2,0,\lambda_2,0)\frac{1}{\omega - (\epsilon_0 - t + \frac{c+U}{2}) + i\eta}, \tag{46}$$

where

$$\lambda_1 = 1 + \frac{(A+1)^2}{a^2}, \qquad \lambda_2 = 1 + \frac{(A-1)^2}{a^2}. \tag{47}$$

We observe that the on-site interaction $U$ only influences the positions of the poles of $G_3^h$ but not their amplitudes. Instead, for $G_3^e$ the interaction strength affects both the poles and their amplitudes. In the non-interacting limit these expressions become

$$G_{03}^h(\omega) = \mathrm{diag}(0,1,0,1)\frac{1}{\omega - (\epsilon_0 - t) - i\eta} + \mathrm{diag}(1,0,1,0)\frac{1}{\omega - (\epsilon_0 - 3t) - i\eta}, \tag{48}$$

$$G_{03}^e(\omega) = \mathrm{diag}(0,1,0,1)\frac{1}{\omega - (\epsilon_0 + 3t) + i\eta} + \mathrm{diag}(2,0,2,0)\frac{1}{\omega - (\epsilon_0 + t) + i\eta}. \tag{49}$$

As was the case at 1/4 filling, $G_{03}$ contains information about satellites, i.e., the terms corresponding to the poles at $\epsilon_0 \pm 3t$, with non-vanishing amplitudes. As mentioned before, we could treat separately the addition and removal parts of $G_3^{e+h}$. However, this would mean that we have to solve two Dyson-like equations, one for $G_3^e$ and $G_3^h$.

In order to use a single Dyson equation, i.e., Eq. (18), we need the total $G_3^{e+h}$. The expression for $G_3^{e+h}$ in its diagonal basis is given by

$$G_3^{e+h}(\omega) = \mathrm{diag}(\xi_1,\xi_2,\xi_3,\xi_4,\xi_1,\xi_2,\xi_3,\xi_4), \tag{50}$$

where

$$\xi_{1,2} = \frac{1}{2}\frac{1}{\omega-(\epsilon_0+t+\frac{U-c}{2})-i\eta} + \frac{1}{2}\frac{\lambda_1}{\omega-(\epsilon_0+t+\frac{U+c}{2})+i\eta} \mp \left(\frac{1}{4}\frac{1}{(\omega-(\epsilon_0+t+\frac{U-c}{2})-i\eta)^2}\right.$$
$$\left. + \frac{1}{4}\frac{\lambda_1^2}{(\omega-(\epsilon_0+t+\frac{U+c}{2})+i\eta)^2} - \frac{1/2+A/a^2+4A^2/a^4}{(\omega-(\epsilon_0+t+\frac{U+c}{2})+i\eta)(\omega-(\epsilon_0+t+\frac{U-c}{2})-i\eta)}\right)^{1/2},$$

$$\xi_{3,4} = \frac{1}{2}\frac{1}{\omega-(\epsilon_0-t+\frac{U-c}{2})-i\eta} + \frac{1}{2}\frac{\lambda_2}{\omega-(\epsilon_0-t+\frac{U+c}{2})+i\eta} \mp \left(\frac{1}{4}\frac{1}{(\omega-(\epsilon_0-t+\frac{U-c}{2})-i\eta)^2}\right.$$
$$\left. + \frac{1}{4}\frac{\lambda_2^2}{(\omega-(\epsilon_0-t+\frac{U+c}{2})+i\eta)^2} - \frac{1/2-A/a^2+4A^2/a^4}{(\omega-(\epsilon_0-t+\frac{U+c}{2})+i\eta)(\omega-(\epsilon_0-t+\frac{U-c}{2})-i\eta)}\right)^{1/2}. \quad (51)$$

We note that it is not the sum of Equations (45) and (46), because $G_3^e$ and $G_3^h$ are diagonal in a different basis. We thus find four distinct eigenvalues, each one with multiplicity 2 since the matrix is block diagonal with respect to the spin of the added particle (electron or hole).

Despite the complexity of the eigenvalues in Eq. (51), in the non-interacting limit $G_3^{e+h}$ becomes very simple, i.e.,

$$G_{03}^{e+h}(\omega) = \text{diag}(\xi_1^0, \xi_2^0, \xi_3^0, \xi_4^0, \xi_1^0, \xi_2^0, \xi_3^0, \xi_4^0), \quad (52)$$

where

$$\xi_1^0 = \frac{1}{\omega-(\epsilon_0-t)-i\eta},$$
$$\xi_2^0 = \frac{1}{\omega-(\epsilon_0+3t)+i\eta},$$
$$\xi_3^0 = \frac{1}{\omega-(\epsilon_0-3t)-i\eta},$$
$$\xi_4^0 = \frac{2}{\omega-(\epsilon_0+t)+i\eta}, \quad (53)$$

which shows that, as expected, the amplitudes related to satellites are non-zero also in the non-interacting case.

From eqs. (50) and (52) we can find an analytical expression for the 3-SE by using Eq. (19). However, we will not report the explicit expression of the 3-SE here because it would take up too much space. The main difference with the exact 3-SE of the Hubbard dimer at 1/4 filling is that at 1/2 filling the 3-SE is frequency dependent. Therefore, this case is a good test to check the accuracy of the static approximation given in Eq. (25).

For comparison, we also report $G_1$ which, in the bonding/anti-bonding basis, reads

$$G_1(\omega) = \text{diag}(G_1'(\omega), G_1'(\omega), G_1''(\omega), G_1''(\omega)), \quad (54)$$

where

$$G_1'(\omega) = \frac{1}{a^2}\left[\frac{(1+A)^2}{\omega-(\epsilon_0+\frac{U+c}{2}+t)+i\eta} + \frac{(1-A)^2}{\omega-(\epsilon_0+\frac{U-c}{2}+t)-i\eta}\right],$$
$$G_1''(\omega) = \frac{1}{a^2}\left[\frac{(1-A)^2}{\omega-(\epsilon_0+\frac{U+c}{2}-t)+i\eta} + \frac{(1+A)^2}{\omega-(\epsilon_0+\frac{U-c}{2}-t)-i\eta}\right]. \quad (55)$$

This matrix is block diagonal in the spin.

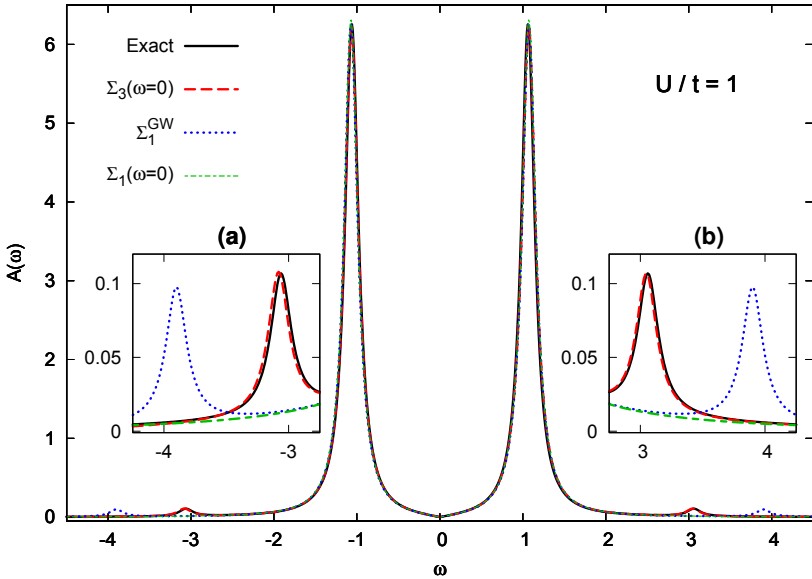

Figure 4: The spectral function of the Hubbard dimer at 1/2 filling at weak interaction ($U/t = 1$) obtained with various levels of theory. Exact result (black solid line); the 1-GF obtained from the exact static 3-SE (red dashed line); the *GW* approximation (blue dotted line); the exact static 1-GF (green dashed line). The outer peaks are the satellites. (a) zoom of the removal satellite; (b) zoom of the addition satellite. The spectra correspond to $\epsilon_0 = -U/2$ which guarantees the particle-hole symmetry.

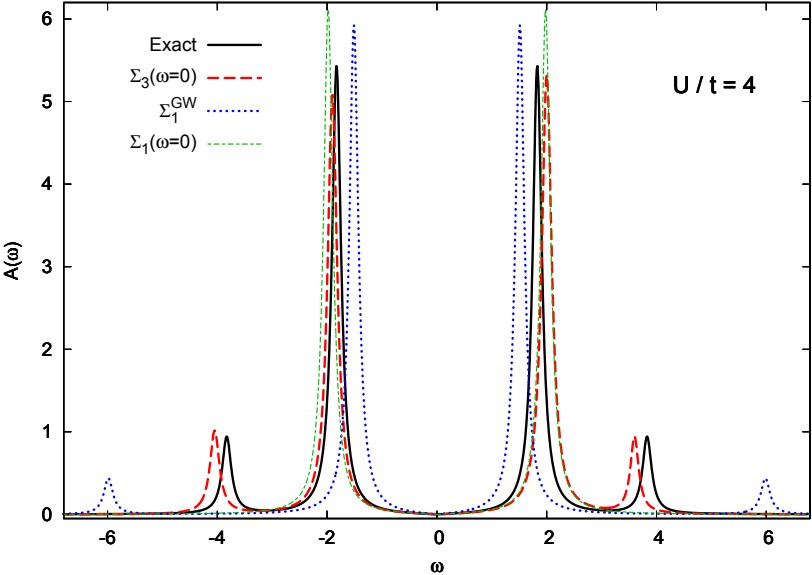

Figure 5: The spectral function of the Hubbard dimer at 1/2 filling at strong interaction ($U/t = 4$) obtained with various levels of theory. Exact result (black solid line); the 1-GF obtained from the exact static 3-SE (red dashed line); the *GW* approximation (blue dotted line); the exact static 1-GF (green dashed line). The outer peaks are the satellites. The spectra correspond to $\epsilon_0 = -U/2$ which guarantees the particle-hole symmetry.

In Figures 4 and 5 we compare the exact spectral function for the Hubbard dimer at 1/2 filling for $U/t = 1$ and $U/t = 4$, respectively, to the spectral functions obtained with the following three approximations: 1) the calculation of $G_3^{e+h}$ using the exact static approximation for the 3-SE according to Eq. (25) ($\Sigma_3(\omega = 0)$) followed by the application of eqs. (34) and (35), 2) the calculation of the 1-GF using the exact static approximation of the 1-SE ($\Sigma_1(\omega = 0)$) and 3) the *GW* approximation [36]. At weak interaction (Figure 4) the quasiparticle peaks are very well described by all the approximations considered. On the contrary, the agreement with the exact result for the satellites are very good only with the static approximation to the exact 3-SE proposed in this paper. The *GW* approximation overestimates the energy difference with the nearest quasiparticle energy, while in the spectral function obtained from static approximation to the 1-SE the satellite amplitudes are not present at all. At strong interaction (Figure 5) the quasiparticle energies are still well described by the two static approximations, although the gap between the two quasiparticle energies is slightly overestimated. Instead, the *GW* approximation significantly underestimates this gap. The satellites are only well described by the static approximation to the 3-SE, as they are absent in the spectral function obtained from the static approximation to the 1-SE, while the *GW* approximation completely fails to reproduce the positions of the satellites and severely underestimates its amplitudes.

## 4 Conclusions and Outlook

We have given a proof of principle that the three-body Green's function is a promising quantity to describe photoemission spectra, especially for correlated systems in which satellites play an important role. In particular, we have shown that $G_3^{e+h}$ which is the sum of the electron-hole-hole and electron-electron-hole parts of the three-body Green's function, contains all the necessary information to describe the spectral function. Indeed, we have shown explicitly how one can retrieve the one-body Green's function $G_1$ from $G_3^{e+h}$. We have demonstrated that an important advantage of $G_3^{e+h}$ with respect to $G_1$ is that its non-interacting counterpart $G_{03}^{e+h}$ already contains information about satellites. Therefore, even when the corresponding three-body self-energy, which relates $G_{03}^{e+h}$ to $G_3^{e+h}$, is chosen to be static the spectral function still contains satellite structures. This should be compared to the spectral function obtained through a static one-body self-energy in which such structures are completely absent. An advantage of a static self-energy is that the self-consistent solution of the Dyson equation can be readily implemented. We have illustrated the above principles by studying the spectral function of the symmetric Hubbard dimer at 1/4 and 1/2 filling. For this model system the static approximation to the exact three-body self-energy yields excellent results even at strong correlation.

For the specific case of the Hubbard dimer we were able to obtain the exact $G_3^{e+h}$. Therefore we could obtain an exact three-body self-energy by solving a Dyson equation. However, in general, the exact three-body self-energy is unknown. Therefore, our next goal is to derive a general static approximation for the three-body self-energy. This could be achieved, for example, by using the equation of motion for $G_3^{e+h}$ [38] along the same lines as has been done for $G_1$ or by using a similar strategy as in Ref. [27], where a practical scheme to calculate $G_3$ for the description of Auger spectra is proposed. Finally, we note that, due to its three-body nature, the calculation of $G_3^{e+h}$ could be a computational challenge for real materials. However, one-body Green's functions of molecules and solids are being calculated for almost 40 years now [39, 40], while two-body Green's functions, or related quantities, of real systems are being calculated for more than 20 years now [41–43]. Moreover, those calculations are nowadays routinely performed for systems containing many elec-

trons. Therefore, we think that it is timely to explore the numerical calculation of three-body Green's functions. Moreover, we can reduce the numerical cost of the calculations by applying a diagonal approximation to the three-body self-energy, in similar manner as is often done for the one-body self-energy, to calculate only the poles of $G_3^{e+h}$. While a static one-body self-energy can only yield poles that correspond to quasi-particles, a diagonal static three-body self-energy would yield the poles corresponding to both quasi-particles and satellites.

Finally, the three-body Green's functions could be used to simulate other types of excitations, e.g., trions [44–47], or to compute time- and angle-resolved photoemission spectra of systems with excitons in which at least three particles (the electron-hole pair (exciton) already present in the system and the additional particle added to the system) are involved [48, 49].

## Acknowledgments

We thank the French Agence Nationale de la Recherche (ANR) for financial support (Grant Agreements No. ANR-18-CE30- 0025 and ANR-19-CE30-0011).

## A Derivation of the spectral representation of $G_3$

Here we derive Eq. (2) starting from the definition of $G_3$ given by Eq. (1). We follow a similar procedure that Csanak *et al.* [35] used to find the e-h/h-e part of the two-particle Green's function. We start by considering two different time orderings which we will discuss in the following,

*Case 1:* $t_1, t_2, t_{3'} > t_3, t_{1'}, t_{2'}$
We set $t_1, t_2, t_{3'} > t_3, t_{1'}, t_{2'}$ without fixing the order of $t_1, t_2$ and $t_{3'}$, and of $t_3, t_{1'}$ and $t_{2'}$. Then, for this time order we have

$$
\begin{aligned}
G_3^e(1,2,3,1',2',3') &= -i\langle\Psi_0^N|T[\hat{\psi}_H(1)\hat{\psi}_H(2)\hat{\psi}_H^\dagger(3')]T[\hat{\psi}_H(3)\hat{\psi}_H^\dagger(2')\hat{\psi}_H^\dagger(1')]|\Psi_0^N\rangle \\
&= -i\sum_n\langle\Psi_0^N|T[\hat{\psi}_H(1)\hat{\psi}_H(2)\hat{\psi}_H^\dagger(3')]|\Psi_n^{N+1}\rangle\langle\Psi_n^{N+1}|T[\hat{\psi}_H(3)\hat{\psi}_H^\dagger(2')\hat{\psi}_H^\dagger(1')]|\Psi_0^N\rangle \\
&= -i\sum_n \chi_n(1,2,3')\tilde{\chi}_n(3,2',1') = i\sum_n \chi_n(1,2,3')\tilde{\chi}_n(1',2',3),
\end{aligned}
\tag{56}
$$

where we used the closure relation in Fock space. The electron-electron-hole amplitudes have been defined as

$$
\begin{aligned}
\chi_n(1,2,3') &= \langle\Psi_0^N|T[\hat{\psi}_H(1)\hat{\psi}_H(2)\hat{\psi}_H^\dagger(3')]|\Psi_n^{N+1}\rangle, \\
\tilde{\chi}_n(1',2',3) &= \langle\Psi_n^{N+1}|T[\hat{\psi}_H^\dagger(1')\hat{\psi}_H^\dagger(2')\hat{\psi}_H(3)]|\Psi_0^N\rangle.
\end{aligned}
\tag{57}
$$

Making explicit the times in the Heisenberg representation of the field operators, these amplitudes can be rewritten as

$$
\begin{aligned}
\chi_n(1,2,3') &= \exp[i/3(t_1+t_2+t_{3'})(E_0^N-E_n^{N+1})]\, X_n(x_1,x_2,x_{3'};\tau_{12},\tau_{23'}), \\
\tilde{\chi}_n(1',2',3) &= \exp[-i/3(t_{1'}+t_{2'}+t_3)(E_0^N-E_n^{N+1})]\, \tilde{X}_n(x_{1'},x_{2'},x_3;\tau_{1'2'},\tau_{2'3}),
\end{aligned}
\tag{58}
$$

where X and $\tilde{X}$ are defined by

$$
\begin{aligned}
X_n(x_1, x_2, x_{3'}; \tau_{12}, \tau_{23'}) = \sum_{i \neq j \neq k = 1, 2, 3'} (-1)^P \theta(\tau_{ij}) \theta(\tau_{jk}) \exp[\frac{i}{3}(E_0^N(2\tau_{ij} + \tau_{jk}) \\
+ E_n^{N+1}(2\tau_{jk} + \tau_{ij}))] \langle \Psi_0^N | \Upsilon(x_i) e^{-iH\tau_{ij}} \Upsilon(x_j) e^{-iH\tau_{jk}} \Upsilon(x_k) | \Psi_n^{N+1} \rangle, \quad (59)
\end{aligned}
$$

$$
\begin{aligned}
\tilde{X}_n(x_{1'}, x_{2'}, x_3; \tau_{1'2'}, \tau_{2'3}) = \sum_{i \neq j \neq k = 1', 2', 3} (-1)^P \theta(\tau_{ij}) \theta(\tau_{jk}) \exp[\frac{i}{3}(E_n^{N+1}(2\tau_{ij} + \tau_{jk}) \\
+ E_0^N(2\tau_{jk} + \tau_{ij}))] \langle \Psi_n^{N+1} | \Upsilon(x_i) e^{-iH\tau_{ij}} \Upsilon(x_j) e^{-iH\tau_{jk}} \Upsilon(x_k) | \Psi_0^N \rangle, \quad (60)
\end{aligned}
$$

where $P$ is the number of permutations with respect to the initial order $i = 1$, $j = 2$, $k = 3'$ or $i = 1'$, $j = 2'$, $k = 3$. Finally, $\Upsilon(x_i)$ is given by

$$
\Upsilon(x_i) = \begin{cases} \hat{\psi}(x_i) & \text{if} \quad i = 1, 2, 3 \\ \hat{\psi}^\dagger(x_i) & \text{if} \quad i = 1', 2', 3'. \end{cases} \quad (61)
$$

Using Eq. (56), this yields

$$
\begin{aligned}
G_3^e(1, 2, 3, 1', 2', 3') = i \sum_n \exp[i\tau(E_0^N - E_n^{N+1})] X_n(x_1, x_2, x_{3'}; \tau_{12}, \tau_{23'}) \\
\times \tilde{X}_n(x_{1'}, x_{2'}, x_3; \tau_{1'2'}, \tau_{2'3}), \quad (62)
\end{aligned}
$$

where we defined

$$
\tau = \frac{1}{3}(t_1 + t_2 + t_{3'}) - \frac{1}{3}(t_3 + t_{1'} + t_{2'}), \qquad \tau_{ij} = t_i - t_j. \quad (63)
$$

The important point is that Eq. (62) depends on $\tau$ only through an exponential factor in which it multiplies the electron addition energies. As a consequence, the Fourier transform of $G_3^e(\tau)$ has poles at these energies, the calculation of which are, along with the corresponding amplitudes, the main objective of this work. The only other time ordering that yields a $G_3(\tau)$ of which the time $\tau$ is completely factorable from the matrix elements is $t_3, t_{1'}, t_{2'} > t_1, t_2, t_{3'}$. We will discuss it in the following subsection.

*Case 2: $t_3, t_{1'}, t_{2'} > t_1, t_2, t_{3'}$*
For this order of the times we obtain

$$
\begin{aligned}
G_3^h(1, 2, 3, 1', 2', 3') &= i \langle \Psi_0^N | T[\hat{\psi}_H(3) \hat{\psi}_H^\dagger(2') \hat{\psi}_H^\dagger(1')] T[\hat{\psi}_H(1) \hat{\psi}_H(2) \hat{\psi}_H^\dagger(3')] | \Psi_0^N \rangle \\
&= i \sum_n \langle \Psi_0^N | T[\hat{\psi}_H(3) \hat{\psi}_H^\dagger(2') \hat{\psi}_H^\dagger(1')] | \Psi_n^{N-1} \rangle \langle \Psi_n^{N-1} | T[\hat{\psi}_H(1) \hat{\psi}_H(2) \hat{\psi}_H^\dagger(3')] | \Psi_0^N \rangle \\
&= i \sum_n \tilde{\zeta}_n(3, 2', 1') \zeta_n(1, 2, 3') = -i \sum_n \tilde{\zeta}_n(1', 2', 3) \zeta_n(1, 2, 3'), \quad (64)
\end{aligned}
$$

where again we used the completeness of the Fock space. The hole-hole-electron amplitudes have been defined as

$$
\begin{aligned}
\zeta_n(1, 2, 3') &= \langle \Psi_n^{N-1} | T[\hat{\psi}_H(1) \hat{\psi}_H(2) \hat{\psi}_H^\dagger(3')] | \Psi_0^N \rangle, \\
\tilde{\zeta}_n(1', 2', 3) &= \langle \Psi_0^N | T[\hat{\psi}_H^\dagger(1') \hat{\psi}_H^\dagger(2') \hat{\psi}_H(3)] | \Psi_n^{N-1} \rangle. \quad (65)
\end{aligned}
$$

As before, we make explicit the times in the Heisenberg representation of the field operators, arriving at

$$
\begin{aligned}
\zeta_n(1, 2, 3') &= \exp[-i/3(t_1 + t_2 + t_{3'})(E_0^N - E_n^{N-1})] Z_n(x_1, x_2, x_{3'}; \tau_{12}, \tau_{23'}), \\
\tilde{\zeta}_n(1', 2', 3) &= \exp[i/3(t_{1'} + t_{2'} + t_3)(E_0^N - E_n^{N-1})] \tilde{Z}_n(x_{1'}, x_{2'}, x_3; \tau_{1'2'}, \tau_{2'3}), \quad (66)
\end{aligned}
$$

where Z and $\tilde{Z}$ are defined by

$$
\begin{aligned}
Z_n(x_1, x_2, x_{3'}; \tau_{12}, \tau_{23'}) = \sum_{i \neq j \neq k = 1,2,3'} (-1)^P \theta(\tau_{ij}) \theta(\tau_{jk}) \exp[\frac{i}{3}(E_0^N(2\tau_{jk} + \tau_{ij}) \\
+ E_n^{N-1}(2\tau_{ij} + \tau_{jk}))] \langle \Psi_n^{N-1}| \Upsilon(x_i) e^{-iH\tau_{ij}} \Upsilon(x_j) e^{-iH\tau_{jk}} \Upsilon(x_k)|\Psi_0^N\rangle,
\end{aligned}
\tag{67}
$$

$$
\begin{aligned}
\tilde{Z}_n(x_{1'}, x_{2'}, x_3; \tau_{1'2'}, \tau_{2'3}) = \sum_{i \neq j \neq k = 1',2',3} (-1)^P \theta(\tau_{ij}) \theta(\tau_{jk}) \exp[\frac{i}{3}(E_n^{N-1}(2\tau_{jk} + \tau_{ij}) \\
+ E_0^N(2\tau_{ij} + \tau_{jk}))] \langle \Psi_0^N| \Upsilon(x_i) e^{-iH\tau_{ij}} \Upsilon(x_j) e^{-iH\tau_{jk}} \Upsilon(x_k)|\Psi_n^{N-1}\rangle.
\end{aligned}
\tag{68}
$$

Using (64), this yields

$$
\begin{aligned}
G_3^h(1,2,3,1',2',3') = -i \sum_n \exp[-i\tau(E_0^N - E_n^{N-1})] \tilde{Z}_n(x_{1'}, x_{2'}, x_3; \tau_{1'2'}, \tau_{2'3}) \\
\times Z_n(x_1, x_2, x_{3'}; \tau_{12}, \tau_{23'}).
\end{aligned}
\tag{69}
$$

In this case the time difference $\tau$ multiplies electron removal energies and, therefore, the Fourier transform of $G_3^h(\tau)$ has poles at these energies. By a similar analysis one can show that the other time orderings do not have factorizable exponential in terms of $\tau$.

Therefore, we can write $G_3$ as follows

$$
\begin{aligned}
G_3(1,2,3,1',2',3') = G_3^e(1,2,3,1',2',3') \theta(\tau + F(\tau_{12}, \tau_{3'1}, \tau_{1'2'}, \tau_{31'})) \\
+ G_3^h(1,2,3,1',2',3') \theta(-\tau + F(\tau_{1'2'}, \tau_{31'}, \tau_{12}, \tau_{3'1})) \\
+ \text{other orderings},
\end{aligned}
\tag{70}
$$

where the time orderings of the two cases described above are ensured by the Heaviside functions, and $F$ is defined as

$$
F(\tau_{12}, \tau_{3'1}, \tau_{1'2'}, \tau_{31'}) = \sum_{i \neq j \neq k = 1,2,3'} \frac{1}{3}(\tau_{ij} - \tau_{ki}) \theta(\tau_{jk}) \theta(\tau_{ki}) - \sum_{i \neq j \neq k = 1',2',3} \frac{1}{3}(\tau_{ij} - \tau_{ki}) \theta(\tau_{jk}) \theta(\tau_{ij}).
\tag{71}
$$

The term *other orderings* refers to all the other possible time permutations present in the initial definition of $G_3$ given in Eq. (1). Since it is impossible to factorize an exponential of the form $\exp[\pm i\tau(E_0^N - E_n^{N\pm1})]$ in any of the terms in *other orderings*, when we perform the Fourier transform respect to $\tau$ all these terms are nonsingular at frequencies equal to electron removal or addition energies.

For this reason, we define the first two terms on the right-hand side of Eq. (70) as the electron-electron-hole/hole-hole-electron Green's function $G_3^{e+h}$,

$$
\begin{aligned}
G_3^{e+h}(1,2,3,1',2',3') = i \sum_n X_n(x_1, x_2, x_{3'}; \tau_{12}, \tau_{23'}) \tilde{X}_n(x_{1'}, x_{2'}, x_3; \tau_{1'2'}, \tau_{2'3}) \\
\times \exp[i\tau(E_0^N - E_n^{N+1})] \theta(\tau + F(\tau_{12}, \tau_{3'1}, \tau_{1'2'}, \tau_{31'})) \\
-i \sum_n \tilde{Z}_n(x_{1'}, x_{2'}, x_3; \tau_{1'2'}, \tau_{2'3}) Z_n(x_1, x_2, x_{3'}; \tau_{12}, \tau_{23'}) \\
\times \exp[-i\tau(E_0^N - E_n^{N-1})] \theta(-\tau + F(\tau_{1'2'}, \tau_{31'}, \tau_{12}, \tau_{3'1})),
\end{aligned}
\tag{72}
$$

which is equal to Eq. (2). The spectral representation of $G_3$ is obtained by Fourier transforming with respect to $\tau$ which yields

$$
\begin{aligned}
G_3^{e+h}(x_1,x_2,x_3,x_{1'},x_{2'},x_{3'}; \tau_{12}, \tau_{2,3'}, \tau_{1'2'}, \tau_{2'3}, \omega) = G_3^e(x_1,x_2,x_3,x_{1'},x_{2'},x_{3'}; \tau_{12}, \tau_{2,3'}, \tau_{1'2'}, \tau_{2'3}, \omega) \\
+ G_3^h(x_1, x_2, x_3, x_{1'}, x_{2'}, x_{3'}; \tau_{12}, \tau_{2,3'}, \tau_{1'2'}, \tau_{2'3}, \omega)
\end{aligned}
\tag{73}
$$

where

$$G_3^e(x_1, x_2, x_3, x_{1'}, x_{2'}, x_{3'}; \tau_{12}, \tau_{2,3'}, \tau_{1'2'}, \tau_{2'3}, \omega) \tag{74}$$

$$= -\sum_n e^{-i[\omega - (E_n^{N+1} - E_0^N)]F(\tau_{12}, \tau_{3'1}, \tau_{1'2'}, \tau_{31'})} \frac{X_n(x_1, x_2, x_{3'}; \tau_{12}, \tau_{23'})\tilde{X}_n(x_{1'}, x_{2'}, x_3; \tau_{1'2'}, \tau_{2'3})}{\omega - (E_n^{N+1} - E_0^N) + i\eta},$$

and

$$G_3^h(x_1, x_2, x_3, x_{1'}, x_{2'}, x_{3'}; \tau_{12}, \tau_{2,3'}, \tau_{1'2'}, \tau_{2'3}, \omega \tag{75}$$

$$= -\sum_n e^{-i[\omega - (E_0^N - E_n^{N-1})]F(\tau_{1'2'}, \tau_{31'}, \tau_{12}, \tau_{3'1})} \frac{\tilde{Z}_n(x_{1'}, x_{2'}, x_3; \tau_{1'2'}, \tau_{2'3})Z_n(x_1, x_2, x_{3'}; \tau_{12}, \tau_{23'})}{\omega - (E_0^N - E_n^{N-1}) - i\eta},$$

which is equal to Eq. (10).

Finally, we emphasize that it is possible to perform a similar derivation for other time orderings, except those corresponding to three electrons or three holes, i.e., $t_1, t_2, t_3 > t_{1'}, t_{2'}, t_{3'}$ and $t_1, t_2, t_3 < t_{1'}, t_{2'}, t_{3'}$. For each choice we can obtain an equation that is similar to Eq. (70), i.e., only two terms are singular at the electron removal and addition energies. Therefore, the final result is independent of this choice.

## B  Recovering $G_1$ from $G_3$

To recover $G_1$ from $G_3$ we have to contract the spin-position variables in the field operators corresponding to the neutral excitation and integrate over the remaining two variables. We thus obtain

$$\int dx_2 dx_3 dx_{2'} dx_{3'} \delta(x_2 - x_{3'})\delta(x_3 - x_{2'})G_3^{e+h}(x_1, x_2, x_3, x_{1'}, x_{2'}, x_{3'}; \omega)$$

$$= \int dx_2 dx_3 G_3^{e+h}(x_1, x_2, x_3, x_{1'}, x_3, x_2; \omega)$$

$$= \int dx_2 dx_3 \sum_n \left[ \frac{\langle\Psi_0^N|\hat{\psi}^\dagger(x_2)\hat{\psi}(x_2)\hat{\psi}(x_1)|\Psi_n^{N+1}\rangle\langle\Psi_n^{N+1}|\hat{\psi}^\dagger(x_{1'})\hat{\psi}^\dagger(x_3)\hat{\psi}(x_3)|\Psi_0^N\rangle}{\omega - (E_n^{N+1} - E_0^N) + i\eta} \right.$$

$$\left. + \frac{\langle\Psi_0^N|\hat{\psi}^\dagger(x_{1'})\hat{\psi}^\dagger(x_3)\hat{\psi}(x_3)|\Psi_n^{N-1}\rangle\langle\Psi_n^{N-1}|\hat{\psi}^\dagger(x_2)\hat{\psi}(x_2)\hat{\psi}(x_1)|\Psi_0^N\rangle}{\omega - (E_0^N - E_n^{N-1}) - i\eta} \right]$$

$$= \sum_n \left[ N^2 \frac{\langle\Psi_0^N|\hat{\psi}(x_1)|\Psi_n^{N+1}\rangle\langle\Psi_n^{N+1}|\hat{\psi}^\dagger(x_{1'})|\Psi_0^N\rangle}{\omega - (E_n^{N+1} - E_0^N) + i\eta} \right.$$

$$\left. + (N-1)^2 \frac{\langle\Psi_0^N|\hat{\psi}^\dagger(x_{1'})|\Psi_n^{N-1}\rangle\langle\Psi_n^{N-1}|\hat{\psi}(x_1)|\Psi_0^N\rangle}{\omega - (E_0^N - E_n^{N-1}) - i\eta} \right]$$

$$= N^2 G_1^e(x_1, x_{1'}; \omega) + (N-1)^2 G_1^h(x_1, x_{1'}; \omega), \tag{76}$$

where $e(h)$ refers to the addition(removal) part of the 1-GF and where we used that

$$\int dx\, \hat{\psi}^\dagger(x)\hat{\psi}(x)|\Psi_n^N\rangle = N|\Psi_n^N\rangle. \tag{77}$$

From the relation $G_3^{e+h} = G_3^e + G_3^h$ one can then easily obtain Eqs. (17).

## C   Inversion of the Dyson equation

In order to obtain Eq. (19) from the Dyson equation (18) we define the inverse of the three-body Green's function $G_3^{e+h}$ according to

$$G_3^{e+h}(x_1,x_2,x_3,x_{1'},x_{2'},x_{3'};\omega)[G_3^{e+h}]^{-1}(x_{1'},x_{2'},x_{6'},x_4,x_5,x_3;\omega)=\delta(x_1-x_4)\delta(x_2-x_5)\delta(x_{3'}-x_{6'}),  \quad (78)$$

$$[G_3^{e+h}]^{-1}(x_{4'},x_{5'},x_{3'},x_1,x_2,x_6;\omega)G_3^{e+h}(x_1,x_2,x_3,x_{1'},x_{2'},x_{3'};\omega)=\delta(x_{1'}-x_{4'})\delta(x_{2'}-x_{5'})\delta(x_3-x_6),  \quad (79)$$

where repeated variables are integrated over. Applying $[G_3^{e+h}]^{-1}(x_{1'},x_{2'},x_{9'},x_7,x_8,x_3;\omega)$ on the right and $[G_{03}^{e+h}]^{-1}(x_{7'},x_{8'},x_{3'},x_1,x_2,x_9)$ on the left in Eq. (18) and integrating over the coordinate $x_1,x_2,x_3,x_{1'},x_{2'},x_{3'}$ we obtain Eq. (19)

## D   Eigenvalues and eigenvectors of the symmetric Hubbard dimer

To keep the paper self-contained we report here the results obtained in Refs. [36] for the eigensystem of the Hamiltonian in Eq. (36). These results will be used in Appendices E and F. The eigenstates of the system are linear combinations of Slater determinants, which are denoted by the kets $|1\ 2\rangle$, with occupations of the sites 1, 2 given by 0, ↑, ↓, ↑↓. In Tables 1 to 3 we report the eigenvalues and the coefficients of the eigenvectors for the symmetric Hubbard dimer for $N = 1, 2, 3$ respectively.

## E   Diagonal $G_3^{e+h}$ for Hubbard dimer at 1/4 filling

In this section we resolve $G_3^e$ (addition part) for the symmetric Hubbard dimer filled with only one electron. We consider the ground state to be the symmetric combination of spin up states, i.e., $|\psi_0^{N=1}\rangle = 1/\sqrt{2}(|\uparrow\ 0\rangle + |0\ \uparrow\rangle)$. In order to build the addition part of $G_3$, it is convenient to list all non-vanishing contributions for $\hat{c}_m^\dagger\hat{c}_o^\dagger\hat{c}_k|\psi_0^{N=1}\rangle$ (see Eq. (33)). They

Table 1: Eigenvalues and coefficients of the symmetric Hubbard dimer for $N = 1$

| $E_i$ | $\|\uparrow\ 0\rangle$ | $\|\downarrow\ 0\rangle$ | $\|0\ \uparrow\rangle$ | $\|0\ \downarrow\rangle$ |
|---|---|---|---|---|
| $\epsilon_0 - t$ | 0 | $1/\sqrt{2}$ | 0 | $1/\sqrt{2}$ |
| $\epsilon_0 - t$ | $1/\sqrt{2}$ | 0 | $1/\sqrt{2}$ | 0 |
| $\epsilon_0 + t$ | 0 | $1/\sqrt{2}$ | 0 | $-1/\sqrt{2}$ |
| $\epsilon_0 + t$ | $1/\sqrt{2}$ | 0 | $-1/\sqrt{2}$ | 0 |

Table 2: Eigenvalues and coefficients of the symmetric Hubbard dimer for $N = 2$

| $E_i$ | $\|\uparrow\ \downarrow\rangle$ | $\|\downarrow\ \uparrow\rangle$ | $\|\uparrow\ \uparrow\rangle$ | $\|\downarrow\ \downarrow\rangle$ | $\|\uparrow\downarrow\ 0\rangle$ | $\|0\ \uparrow\downarrow\rangle$ |
|---|---|---|---|---|---|---|
| $2\epsilon_0 + (U-c)/2$ | $-\frac{A}{a}$ | $\frac{A}{a}$ | 0 | 0 | $1/a$ | $1/a$ |
| $2\epsilon_0 + (U+c)/2$ | $-\frac{B}{b}$ | $\frac{B}{b}$ | 0 | 0 | $1/b$ | $1/b$ |
| $2\epsilon_0 + U$ | 0 | 0 | 0 | 0 | $-1/\sqrt{2}$ | $1/\sqrt{2}$ |
| $2\epsilon_0$ | 0 | 0 | 0 | 1 | 0 | 0 |
| $2\epsilon_0$ | 0 | 0 | 1 | 0 | 0 | 0 |
| $2\epsilon_0$ | $1/\sqrt{2}$ | $1/\sqrt{2}$ | 0 | 0 | 0 | 0 |

Table 3: Eigenvalues and coefficients of the symmetric Hubbard dimer for $N = 3$

| $E_i$ | $|\uparrow\ \uparrow\downarrow\rangle$ | $|\downarrow\ \uparrow\downarrow\rangle$ | $|\uparrow\downarrow\ \uparrow\rangle$ | $|\uparrow\downarrow\ \downarrow\rangle$ |
|---|---|---|---|---|
| $3\epsilon_0+U-t$ | $0$ | $-1/\sqrt{2}$ | $0$ | $1/\sqrt{2}$ |
| $3\epsilon_0+U-t$ | $-1/\sqrt{2}$ | $0$ | $1/\sqrt{2}$ | $0$ |
| $3\epsilon_0+U+t$ | $0$ | $1/\sqrt{2}$ | $0$ | $1/\sqrt{2}$ |
| $3\epsilon_0+U+t$ | $1/\sqrt{2}$ | $0$ | $1/\sqrt{2}$ | $0$ |

are given by:

1) $\hat{c}_{1\uparrow}^\dagger \hat{c}_{2\uparrow}^\dagger \hat{c}_{2\uparrow} |\psi_0^{N=1}\rangle = -\hat{c}_{2\uparrow}^\dagger \hat{c}_{1\uparrow}^\dagger \hat{c}_{1\uparrow} |\psi_0^{N=1}\rangle = \frac{1}{\sqrt{2}} |\uparrow; \uparrow\rangle$,

2) $\hat{c}_{2\downarrow}^\dagger \hat{c}_{1\uparrow}^\dagger \hat{c}_{1\uparrow} |\psi_0^{N=1}\rangle = \hat{c}_{2\downarrow}^\dagger \hat{c}_{1\uparrow}^\dagger \hat{c}_{2\uparrow} |\psi_0^{N=1}\rangle = -\frac{1}{\sqrt{2}} |\uparrow; \downarrow\rangle$,

3) $\hat{c}_{1\downarrow}^\dagger \hat{c}_{2\uparrow}^\dagger \hat{c}_{1\uparrow} |\psi_0^{N=1}\rangle = \hat{c}_{1\downarrow}^\dagger \hat{c}_{2\uparrow}^\dagger \hat{c}_{2\uparrow} |\psi_0^{N=1}\rangle = \frac{1}{\sqrt{2}} |\downarrow; \uparrow\rangle$,

4) $\hat{c}_{1\downarrow}^\dagger \hat{c}_{1\uparrow}^\dagger \hat{c}_{1\uparrow} |\psi_0^{N=1}\rangle = \hat{c}_{1\downarrow}^\dagger \hat{c}_{1\uparrow}^\dagger \hat{c}_{2\uparrow} |\psi_0^{N=1}\rangle = -\frac{1}{\sqrt{2}} |\uparrow\downarrow; 0\rangle$,

5) $\hat{c}_{2\downarrow}^\dagger \hat{c}_{2\uparrow}^\dagger \hat{c}_{1\uparrow} |\psi_0^{N=1}\rangle = \hat{c}_{2\downarrow}^\dagger \hat{c}_{2\uparrow}^\dagger \hat{c}_{2\uparrow} |\psi_0^{N=1}\rangle = -\frac{1}{\sqrt{2}} |0; \uparrow\downarrow\rangle$,

where we considered that the electron added to the system is different from the electron involved in the neutral excitation and that there is no spin flip in the neutral excitation. Note that these simplifications are valid also for the case at one-half filling. Using these states in Eq. (32), together with ground-state energy of the N-electron system, and the eigenstates and eigenvalues of the $(N+1)$-electron system given in Table 2 we arrive at the following matrix form for $G_3^e$,

$$G_{3(ijl;mok)}^e = \begin{pmatrix} G_3' & 0_{1\times 4} & -G_3' & 0_{1\times 4} \\ 0_{4\times 1} & G_{3,4\times 4} & 0_{4\times 1} & G_{3,4\times 4} \\ -G_3' & 0_{1\times 4} & G_3' & 0_{1\times 4} \\ 0_{4\times 1} & G_{3,4\times 4} & 0_{4\times 1} & G_{3,4\times 4} \end{pmatrix}, \tag{80}$$

with

$$G_3' = \frac{1}{\omega - (\epsilon_0 + t) + i\eta}, \tag{81}$$

and

$$\begin{aligned} G_{3,4\times 4} = &\frac{1}{2} \begin{pmatrix} J_{2\times 2} & 0_{2\times 2} \\ 0_{2\times 2} & 0_{2\times 2} \end{pmatrix} \frac{1}{\omega - (\epsilon_0 + t) + i\eta} + \frac{1}{2} \begin{pmatrix} 0_{2\times 2} & 0_{2\times 2} \\ 0_{2\times 2} & J_{2\times 2} \end{pmatrix} \frac{1}{\omega - (\epsilon_0 + U + t) + i\eta} \\ &+ \frac{1}{b^2} \begin{pmatrix} B^2 I_{2\times 2} & -B I_{2\times 2} \\ -B I_{2\times 2} & I_{2\times 2} \end{pmatrix} \frac{1}{\omega - (\epsilon_0 + \frac{U+c}{2} + t) + i\eta} \\ &+ \frac{1}{a^2} \begin{pmatrix} A^2 I_{2\times 2} & -A I_{2\times 2} \\ -A I_{2\times 2} & I_{2\times 2} \end{pmatrix} \frac{1}{\omega - (\epsilon_0 + \frac{U-c}{2} + t) + i\eta}, \end{aligned} \tag{82}$$

where $A$ and $B$ are defined just below Eq. (43), and

$$I_{2\times 2} = \begin{pmatrix} 1 & 1 \\ 1 & 1 \end{pmatrix}, \qquad J_{2\times 2} = \begin{pmatrix} 1 & -1 \\ -1 & 1 \end{pmatrix}. \tag{83}$$

Diagonalization of matrix (80) produces five non-zero eigenvalues, which are reported in Eq. (37).

# F  Diagonal $G_3^{e+h}$ for Hubbard dimer at 1/2 filling

To obtain the exact expression for the addition part of $G_3^{e+h}$ we start by calculating all the non-zero combinations of $\hat{c}_m^\dagger \hat{c}_o^\dagger \hat{c}_k |\psi_0^{N=2}\rangle$. From Table 2 we learn that the ground state is $|\psi_0^{N=2}\rangle = \frac{A}{a}(|\downarrow;\uparrow\rangle - |\uparrow;\downarrow\rangle) + \frac{1}{a}(|0;\uparrow\downarrow\rangle + |\uparrow\downarrow;0\rangle)$. We thus obtain the following non-vanishing contributions,

1) $\hat{c}_{1\downarrow}^\dagger \hat{c}_{2\uparrow}^\dagger \hat{c}_{2\uparrow} |\psi_0^{N=2}\rangle = \frac{1}{a}|\downarrow;\uparrow\downarrow\rangle$,

11) $\hat{c}_{1\uparrow}^\dagger \hat{c}_{2\downarrow}^\dagger \hat{c}_{2\downarrow} |\psi_0^{N=2}\rangle = \frac{1}{a}|\uparrow;\uparrow\downarrow\rangle$,

2) $\hat{c}_{2\downarrow}^\dagger \hat{c}_{1\uparrow}^\dagger \hat{c}_{1\uparrow} |\psi_0^{N=2}\rangle = \frac{1}{a}|\uparrow\downarrow;\downarrow\rangle$,

12) $\hat{c}_{2\uparrow}^\dagger \hat{c}_{1\downarrow}^\dagger \hat{c}_{1\downarrow} |\psi_0^{N=2}\rangle = \frac{1}{a}|\uparrow\downarrow;\uparrow\rangle$,

3) $\hat{c}_{2\downarrow}^\dagger \hat{c}_{2\uparrow}^\dagger \hat{c}_{2\uparrow} |\psi_0^{N=2}\rangle = \frac{A}{a}|\downarrow;\uparrow\downarrow\rangle$,

13) $\hat{c}_{2\uparrow}^\dagger \hat{c}_{2\downarrow}^\dagger \hat{c}_{2\downarrow} |\psi_0^{N=2}\rangle = \frac{A}{a}|\uparrow;\uparrow\downarrow\rangle$,

4) $\hat{c}_{1\downarrow}^\dagger \hat{c}_{1\uparrow}^\dagger \hat{c}_{1\uparrow} |\psi_0^{N=2}\rangle = \frac{A}{a}|\uparrow\downarrow;\downarrow\rangle$,

14) $\hat{c}_{1\uparrow}^\dagger \hat{c}_{1\downarrow}^\dagger \hat{c}_{1\downarrow} |\psi_0^{N=2}\rangle = \frac{A}{a}|\uparrow\downarrow;\uparrow\rangle$,

5) $\hat{c}_{2\downarrow}^\dagger \hat{c}_{2\uparrow}^\dagger \hat{c}_{1\uparrow} |\psi_0^{N=2}\rangle = -\frac{1}{a}|\downarrow;\uparrow\downarrow\rangle$,

15) $\hat{c}_{2\uparrow}^\dagger \hat{c}_{2\downarrow}^\dagger \hat{c}_{1\downarrow} |\psi_0^{N=2}\rangle = -\frac{1}{a}|\uparrow;\uparrow\downarrow\rangle$,

6) $\hat{c}_{1\downarrow}^\dagger \hat{c}_{1\uparrow}^\dagger \hat{c}_{2\uparrow} |\psi_0^{N=2}\rangle = -\frac{1}{a}|\uparrow\downarrow;\downarrow\rangle$,

16) $\hat{c}_{1\uparrow}^\dagger \hat{c}_{1\downarrow}^\dagger \hat{c}_{2\downarrow} |\psi_0^{N=2}\rangle = -\frac{1}{a}|\uparrow\downarrow;\uparrow\rangle$,

7) $\hat{c}_{1\downarrow}^\dagger \hat{c}_{2\uparrow}^\dagger \hat{c}_{1\uparrow} |\psi_0^{N=2}\rangle = -\frac{A}{a}|\downarrow;\uparrow\downarrow\rangle$,

17) $\hat{c}_{1\uparrow}^\dagger \hat{c}_{2\downarrow}^\dagger \hat{c}_{1\downarrow} |\psi_0^{N=2}\rangle = -\frac{A}{a}|\uparrow;\uparrow\downarrow\rangle$,

8) $\hat{c}_{2\downarrow}^\dagger \hat{c}_{1\uparrow}^\dagger \hat{c}_{2\uparrow} |\psi_0^{N=2}\rangle = -\frac{A}{a}|\uparrow\downarrow;\downarrow\rangle$,

18) $\hat{c}_{2\uparrow}^\dagger \hat{c}_{1\downarrow}^\dagger \hat{c}_{2\downarrow} |\psi_0^{N=2}\rangle = -\frac{A}{a}|\uparrow\downarrow;\uparrow\rangle$,

9) $\hat{c}_{2\downarrow}^\dagger \hat{c}_{1\downarrow}^\dagger \hat{c}_{1\downarrow} |\psi_0^{N=2}\rangle = \frac{A}{a}|\downarrow;\uparrow\downarrow\rangle + \frac{1}{a}|\uparrow\downarrow;\downarrow\rangle$,

19) $\hat{c}_{2\uparrow}^\dagger \hat{c}_{1\uparrow}^\dagger \hat{c}_{1\uparrow} |\psi_0^{N=2}\rangle = \frac{A}{a}|\uparrow;\uparrow\downarrow\rangle + \frac{1}{a}|\uparrow\downarrow;\uparrow\rangle$,

10) $\hat{c}_{1\downarrow}^\dagger \hat{c}_{2\downarrow}^\dagger \hat{c}_{2\downarrow} |\psi_0^{N=2}\rangle = \frac{A}{a}|\uparrow\downarrow;\downarrow\rangle + \frac{1}{a}|\downarrow;\uparrow\downarrow\rangle$,

20) $\hat{c}_{1\uparrow}^\dagger \hat{c}_{2\uparrow}^\dagger \hat{c}_{2\uparrow} |\psi_0^{N=2}\rangle = \frac{A}{a}|\uparrow\downarrow;\uparrow\rangle + \frac{1}{a}|\uparrow;\uparrow\downarrow\rangle$.

We notice that all the states in which we add an electron with spin down have as result a state with two spin-down and one spin-up electrons. Instead, if we add an electron with spin up the resulting state has one spin-down and two spin-up electrons. Therefore, the first ten states listed above are orthogonal to the three-electron states (see Table 3) with one spin-down and two-spin up electrons. Similarly, the last ten states listed above are orthogonal to the three-electron states with one spin-up and two spin-down electrons. Therefore, $G_3^e$ can be written as a block-diagonal $20 \times 20$ matrix with two equal $10 \times 10$ blocks (one for each spin channel of the added electron). The $G_3^e$ in the site basis for one of these $10 \times 10$ blocks reads

$$G^e_{3(ijl;mok)} = \tag{84}$$

$$= \frac{1}{2a^2}
\begin{pmatrix}
1 & 1 & A & A & -1 & -1 & -A & -A & D & D \\
1 & 1 & A & A & -1 & -1 & -A & -A & D & D \\
A & A & A^2 & A^2 & -A & -A & -A^2 & -A^2 & AD & AD \\
A & A & A^2 & A^2 & -A & -A & -A^2 & -A^2 & AD & AD \\
-1 & -1 & -A & -A & 1 & 1 & A & A & -D & -D \\
-1 & -1 & -A & -A & 1 & 1 & A & A & -D & -D \\
-A & -A & -A^2 & -A^2 & A & A & A^2 & A^2 & -AD & -AD \\
-A & -A & -A^2 & -A^2 & A & A & A^2 & A^2 & -AD & -AD \\
D & D & AD & AD & -D & -D & -AD & -AD & D^2 & D^2 \\
D & D & AD & AD & -D & -D & -AD & -AD & D^2 & D^2
\end{pmatrix}
\frac{1}{\omega - (\epsilon_0 + t + \frac{c+U}{2}) + i\eta}$$

$$+\frac{1}{2a^2}\begin{pmatrix} 1 & -1 & A & -A & -1 & 1 & -A & A & C & -C \\ -1 & 1 & -A & A & 1 & -1 & A & -A & -C & C \\ A & -A & A^2 & -A^2 & -A & A & -A^2 & A^2 & AC & -AC \\ -A & A & -A^2 & A^2 & A & -A & A^2 & -A^2 & -AC & AC \\ -1 & 1 & -A & A & 1 & -1 & A & -A & -C & C \\ 1 & -1 & A & -A & -1 & 1 & -A & A & C & -C \\ -A & A & -A^2 & A^2 & A & -A & A^2 & -A^2 & -AC & AC \\ A & -A & A^2 & -A^2 & -A & A & -A^2 & A^2 & AC & -AC \\ C & -C & AC & -AC & -C & C & -AC & AC & C^2 & -C^2 \\ -C & C & -AC & AC & C & -C & AC & -AC & -C^2 & C^2 \end{pmatrix}\frac{1}{\omega-(\epsilon_0-t+\frac{c+U}{2})+i\eta}\,,$$

where we defined $C = A-1$ and $D = A+1$. Both matrices on the right-hand side of Eq. (84) have only one non-zero eigenvalue, namely

$$\lambda_1 = 1+\frac{1}{a^2}D^2\,; \qquad \lambda_2 = 1+\frac{1}{a^2}C^2\,, \tag{85}$$

for the first and second matrices, respectively. $G_3^e$ can therefore be written as the following diagonal matrix

$$G_3^e(\omega) = \mathrm{diag}(0,\lambda_1,0,\lambda_1)\frac{1}{\omega-(\epsilon_0+t+\frac{c+U}{2})+i\eta} + \mathrm{diag}(\lambda_2,0,\lambda_2,0)\frac{1}{\omega-(\epsilon_0-t+\frac{c+U}{2})+i\eta}\,. \tag{86}$$

Let us now consider $G_3^h$ for which we find 16 different non-zero combinations of $c_l^\dagger c_j c_i|\psi_0^{N=2}\rangle$,

1) $\hat{c}_{2\uparrow}^\dagger\hat{c}_{2\uparrow}\hat{c}_{1\downarrow}|\psi_0^{N=2}\rangle = \frac{A}{a}|0;\uparrow\rangle\,,$     11) $\hat{c}_{2\downarrow}^\dagger\hat{c}_{2\downarrow}\hat{c}_{1\uparrow}|\psi_0^{N=2}\rangle = -\frac{A}{a}|0;\downarrow\rangle\,,$

2) $\hat{c}_{1\uparrow}^\dagger\hat{c}_{1\uparrow}\hat{c}_{2\downarrow}|\psi_0^{N=2}\rangle = \frac{A}{a}|\uparrow;0\rangle\,,$     12) $\hat{c}_{1\downarrow}^\dagger\hat{c}_{1\downarrow}\hat{c}_{2\uparrow}|\psi_0^{N=2}\rangle = -\frac{A}{a}|\downarrow;0\rangle\,,$

3) $\hat{c}_{2\uparrow}^\dagger\hat{c}_{2\uparrow}\hat{c}_{2\downarrow}|\psi_0^{N=2}\rangle = -\frac{1}{a}|0;\uparrow\rangle\,,$     13) $\hat{c}_{2\downarrow}^\dagger\hat{c}_{2\downarrow}\hat{c}_{2\uparrow}|\psi_0^{N=2}\rangle = \frac{1}{a}|0;\downarrow\rangle\,,$

4) $\hat{c}_{1\uparrow}^\dagger\hat{c}_{1\uparrow}\hat{c}_{1\downarrow}|\psi_0^{N=2}\rangle = -\frac{1}{a}|\uparrow;0\rangle\,,$     14) $\hat{c}_{1\downarrow}^\dagger\hat{c}_{1\downarrow}\hat{c}_{1\uparrow}|\psi_0^{N=2}\rangle = \frac{1}{a}|\downarrow;0\rangle\,,$

5) $\hat{c}_{2\uparrow}^\dagger\hat{c}_{1\uparrow}\hat{c}_{1\downarrow}|\psi_0^{N=2}\rangle = -\frac{1}{a}|0;\uparrow\rangle\,,$     15) $\hat{c}_{2\downarrow}^\dagger\hat{c}_{1\downarrow}\hat{c}_{1\uparrow}|\psi_0^{N=2}\rangle = \frac{1}{a}|0;\downarrow\rangle\,,$

6) $\hat{c}_{1\uparrow}^\dagger\hat{c}_{2\uparrow}\hat{c}_{2\downarrow}|\psi_0^{N=2}\rangle = -\frac{1}{a}|\uparrow;0\rangle\,,$     16) $\hat{c}_{1\downarrow}^\dagger\hat{c}_{2\downarrow}\hat{c}_{2\uparrow}|\psi_0^{N=2}\rangle = \frac{1}{a}|\downarrow;0\rangle\,,$

7) $\hat{c}_{2\uparrow}^\dagger\hat{c}_{1\uparrow}\hat{c}_{2\downarrow}|\psi_0^{N=2}\rangle = \frac{A}{a}|0;\uparrow\rangle\,,$     17) $\hat{c}_{2\downarrow}^\dagger\hat{c}_{1\downarrow}\hat{c}_{2\uparrow}|\psi_0^{N=2}\rangle = -\frac{A}{a}|0;\downarrow\rangle\,,$

8) $\hat{c}_{1\uparrow}^\dagger\hat{c}_{2\uparrow}\hat{c}_{1\downarrow}|\psi_0^{N=2}\rangle = \frac{A}{a}|\uparrow;0\rangle\,,$     18) $\hat{c}_{1\downarrow}^\dagger\hat{c}_{2\downarrow}\hat{c}_{1\uparrow}|\psi_0^{N=2}\rangle = -\frac{A}{a}|\downarrow;0\rangle\,,$

9) $\hat{c}_{1\downarrow}^\dagger\hat{c}_{1\downarrow}\hat{c}_{2\downarrow}|\psi_0^{N=2}\rangle = 0\,,$     19) $\hat{c}_{1\uparrow}^\dagger\hat{c}_{1\uparrow}\hat{c}_{2\uparrow}|\psi_0^{N=2}\rangle = 0\,,$

10) $\hat{c}_{2\downarrow}^\dagger\hat{c}_{2\downarrow}\hat{c}_{1\downarrow}|\psi_0^{N=2}\rangle = 0\,,$     20) $\hat{c}_{2\uparrow}^\dagger\hat{c}_{2\uparrow}\hat{c}_{1\uparrow}|\psi_0^{N=2}\rangle = 0\,.$

As was the case for $G_3^e$ also $G_3^h$ is block diagonal, with two equal $8\times 8$ blocks. The $G_3^h$ in

the site basis for the one of these $8 \times 8$ block reads

$$G^h_{3(ijl;mok)} = \tag{87}$$

$$= \frac{1}{2a^2}
\begin{pmatrix}
A^2 & A^2 & -A & -A & -A & -A & A^2 & A^2 \\
A^2 & A^2 & -A & -A & -A & -A & A^2 & A^2 \\
-A & -A & 1 & 1 & 1 & 1 & -A & -A \\
-A & -A & 1 & 1 & 1 & 1 & -A & -A \\
-A & -A & 1 & 1 & 1 & 1 & -A & -A \\
-A & -A & 1 & 1 & 1 & 1 & -A & -A \\
A^2 & A^2 & -A & -A & -A & -A & A^2 & A^2 \\
A^2 & A^2 & -A & -A & -A & -A & A^2 & A^2
\end{pmatrix}
\frac{1}{\omega - (\epsilon_0 + t + \frac{U-c}{2}) - i\eta} \tag{88}$$

$$+ \frac{1}{2a^2}
\begin{pmatrix}
A^2 & -A^2 & -A & A & -A & A & A^2 & -A^2 \\
-A^2 & A^2 & A & -A & A & -A & -A^2 & A^2 \\
-A & A & 1 & -1 & 1 & -1 & -A & A \\
A & -A & -1 & 1 & -1 & 1 & A & -A \\
-A & A & 1 & -1 & 1 & -1 & -A & A \\
A & -A & -1 & 1 & -1 & 1 & A & -A \\
A^2 & -A^2 & -A & A & -A & A & A^2 & -A^2 \\
-A^2 & A^2 & A & -A & A & -A & -A^2 & A^2
\end{pmatrix}
\frac{1}{\omega - (\epsilon_0 - t + \frac{U-c}{2}) - i\eta}. \tag{89}$$

The two matrices on the right-hand side of Eq. (87) have only one non-zero eigenvalue of value, which has value one. Therefore, the final expression for $G^h_3(\omega)$ in its diagonal basis can be written as

$$G^h_3(\omega) = \mathrm{diag}(0,1,0,1) \frac{1}{\omega - (\epsilon_0 + t + \frac{U-c}{2}) - i\eta} + \mathrm{diag}(1,0,1,0) \frac{1}{\omega - (\epsilon_0 - t + \frac{U-c}{2}) - i\eta}. \tag{90}$$

So far, we have treated $G^e_3$ and $G^h_3$ separately. However, they are not diagonal in the same basis. The full electron-hole 3-GF $G^{e+h}_3$ in the site basis is obtained by summing Equations (84) and (87). The diagonal $G^{e+h}_3$ is given in Eq. (50).

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
