# Peer review of "Photoemission spectral functions from the three-body Green's function"

_SciPost Physics, doi:SciPost Phys. 12, 093 (2022)_

## Round 1 · Referee Report · Davide Sangalli (Referee 1) · 2021-11-12

Strengths

S1 - defines a new approach to compute photoemission within many-body perturbation theory based on the three-body green function G3. Via G3 a static self-energy could capture satellites and more in general features which would require a dynamical self-energy when using the more standard approach based on G1. This has the potential of solving many open issues in systems where approaches based on G1 fail
S2 - there is a physical and well-done discussion of how time orderings are selected
S3 - the application on the Hubbard dimer, which is exactly solvable, gives some further insight

Weaknesses

There are two aspects that need further clarification
W1 - a satellite in photoemission arise from the interaction between electrons. In the non-interacting limit, only single-particle poles exist. It is ok that the G3 contains extra poles also in the non-interacting case, but such poles should not contribute to the ARPES spectral function, i.e. they should have zero intensity. Only interaction, i.e. a static self-energy, could give finite intensity to these poles. The authors find these poles in the analytical description (eq. 24), and they seem to suggest that they contribute to ARPES also in the non-interacting limit. Indeed there is a pole, which they call satellite, in Fig. 1. This point should be clarified. I think the case is similar to quantum chemistry approaches. When calculations such as CI-SD or CC-SD are performed extra poles (i.e. double excitations) arise. However, they can be seen in photoemission only if they mix via interaction with single excitations (unless one accounts for two-photons processes).
W2 - An important approximation of standard approaches based on G1 (especially in the ab inito community) is to take the diagonal only component of the self-energy, i.e. only the poles are corrected with respect to the zero-order (usually DFT) simulation. G3 has six indexes, and it is not easy to understand where such approximation would enter. This is also related to how demanding would be the inversion of G3 in the QP basis set (appendix C), and which kind of satellites the approach could give. I would expect that, the contraction of the external indexes (indexes im in eqs. 33-34) could give the QP approximation. However, for correlated satellites (plasmons, excitons, magnons, etc .. ) the internal indexes should be allowed to mix. Side comment, something is wrong in the indexes of eq. 30. The Hubbard dimer does not help much here. At least a general discussion in this direction would be useful.

Report

The manuscript is well written and potentially of strong interest for the community. However, the two weaknesses identified should be clarified. In particular W1.

---

## Round 1 · Referee Report · Anonymous (Referee 2) · 2021-11-15

Strengths

The authors consider the calculation of electron-addition spectra (e.g. as measured in photoemission). The exact 1-particle Green's function G1 of many-body perturbation theory is closely related to the measured photoemission spectrum, and it generally includes plasmon satellites and a spectral background, in addition to quasiparticle peaks that are the shifted, renormalized counterparts of the eigenvalues of a corresponding non-interacting Hamiltonian.

  1. Reliably accurate, unbiased methods for calculating G1 (including the non-quasiparticle features) are not available in practice. In this paper the authors consider the 3-particle Green's function G3, on the face of it a much more complex object than G1; however, they show that a rather simple G3 (even one calculated using a omega-dependent self-energy) already contains information about the non-quasiparticle features (e.g. if G3 is used to calculate G1).

  2. The authors test their idea on an extremely simple multi-electron system, the symmetric Hubbard dimer. The Hilbert space of the many-electron wavefunction is very small, which means that the exact G1 and G3 can be calculated, and also a simplified G3 in which the self-energy that appears in the Dyson equation for G3 is constrained to be omega-independent, simplifying the calculations. The G1 calculated from this simplified G3 already contains an impressive account of the non-quasiparticle features.

  3. The paper is clearly written and the analysis of the connections between G3 and G1 in the general case, and for the Hubbard dimer in particular, is impressive.

  4. The paper raises the important and interesting idea of a tradeoff between the degree of the Green's functions used and the level of approximation needed in calculated intermediate quantities.

Weaknesses

The following clarifications are needed, in my view:

  1. In Figs. 1-5 can it be clarified whether a curve labeled G3, or Sigma3, is the 3-particle spectral function or the 1-particle spectral function obtained from a calculation of G3? Hopefully the latter.

  2. Can the actual steps in the calculation of the spectral function using their simplified G3 (static Sigma3) be set out in the language of a general many-electron system and of many-body perturbation theory? E.g. as a flowchart. This would help the reader assess how realistic the Hubbard model is here as a prototype many-electron system. Is Sigma3 spatially non-local? What is the extent of self-consistency imposed by the calculation? To what extent can the recommended procedure be regarded as a true perturbation theory, and if it can, what is the small quantity?

Report

This is an important addition to the literature, even though I suspect an ab initio calculation using a G3 is many years away. I recommend publication in SciPost after clarifications 1-2 above are addressed.

Requested changes

See clarifications above.

---

## Round 2 · Referee Report · Anonymous (Referee 2) · 2021-11-30

Report

The authors have made helpful modifications to their manuscript that provide the clarification that I (and, in my opinion, also the other referee and the Editor) requested.

---

## Round 2 · Referee Report · Davide Sangalli (Referee 1) · 2021-12-3

Report

The authors have addressed the points raised in my review.
In particular for the definition of the ARPES spectral function from G3. The role of the basis set and the related diagonal approximation remains to be further explored. Future applications on real materials could possibly clarify this.

I think the manuscript can be now accepted for publication.

---

## Round 2 · Referee Report · Anonymous (Referee 3) · 2021-12-17

Report

The direct and inverse photoemission intensity is given, within the sudden approximation, by the one-electron removal and addition spectra that are proportional to the imaginary part of the retarded single particle Green function (e.g. Rev. Mod. Phys. 75, 473 2003, Phys. Rev. B 94, 115119 2016). This is the case for both (strongly) interacting and non-interacting/weakly interacting systems. This results stems from a direct calculation of the photoelectron current using scattering theory. Beyond the sudden approximation corrections to the photocurrent appear. Hedin and coworkers (Phys. Rev. B 58 15565 1998) have shown that whereas the sudden approximation includes "intrinsic losses" or satellite structure, adiabatic corrections provide further the "extrinsic losses".

The authors state that the three-body Green function contribute to photoemission intensity, but do not derive this statement from a calculation of the photocurrent. Instead the existing literature (some of which referred to above) seems to agree that within the sudden approximation (and apart from matrix-element effects), the one-particle Green function contains all spectral information relevant for photoemission.

In this context it is not clear how the authors challenge the present status quo and can justify their statement that the three-body Green function is a fundamental quantity to the calculation photoemission spectra. Possibly the implication is that the three-body Green function embodies corrections beyond the sudden approximation. If so, these corrections in terms of the three-body Green function needs to be derived in a mathematically consistent fashion from fundamental considerations on the photocurrent.
  • validity: poor
  • significance: -
  • originality: -
  • clarity: -
  • formatting: -
  • grammar: -

Author:  Arjan Berger  on 2021-12-17  [id 2034]

(in reply to Report 3 on 2021-12-17)
Category:
reply to objection

There seems to be a misunderstanding.
In this work we are always working within the sudden approximation which is the standard in our field. When we write that we use the 3-body Green’s function (3-GF) to calculate photoemission spectra it is implied that we mean photoemission spectra within the sudden approximation. It is not mentioned explicitly since it is standard practice in our field. We will make this point explicit in the second revision of our work in order to avoid any possible misunderstanding by rewriting the following sentence in the introduction

"The main reason is that the one-body Green’s function (1-GF) can be easily linked to photoemission spectra since its poles are the electron removal and addition energies.”

as

"The main reason is that the one-body Green’s function (1-GF) can be easily linked to photoemission spectra (within the sudden approximation) since its poles are the electron removal and addition energies.”

We note that the referee claims that we state that “the three-body Green function contribute to photoemission intensity”.
We want to make clear that nowhere in the paper do we make this statement.

The purpose of this work is completely different.
The final goal is still to calculate the one-body Green’s function (1-GF) since, within the sudden approximation, it indeed has all the required information about photoemission spectra.
However, that is if one has the exact 1-GF.

The main idea of this work is to use the 3-GF to improve the approximations to the 1-GF and, in particular, to capture satellites. This point is explained in detail in the Introduction and Results sections of the paper. In a nutshell, to capture satellites using the standard 1-body approach one requires a dynamical self-energy since the non-interacting 1-GF only contains information about quasi-particles. It is well-known that it is difficult to obtain good dynamical approximations for the self-energy. For example, the GW approximation does not yield very accurate satellites. Instead, when using a 3-body approach, information about satellites is already contained in the non-interacting 3-GF and, therefore, a simpler static 3-body self-energy is sufficient to capture satellites. Once the 3-GF is obtained we contract (according to Eq. (17)) to obtain the 1-GF and therefore the photoemission spectrum including the satellites.
To make this point clearer in the second revision we will modify the following sentence in the Introduction,

"Therefore, we will study here the three-body Green’s function (3-GF) as the fundamental quantity from which to calculate photoemission spectra”

to

“Therefore, we will study here the three-body Green’s function (3-GF) as the fundamental quantity from which to calculate the 1-GF and, hence, photoemission spectra"

Now that this misunderstanding has been cleared up and in view of the two recommendations to publish our work in SciPost Physics already given by the other two referees, we hope that our work can now finally be published in SciPost Physics.

---

## Round 2 · Referee Report · Anonymous (Referee 4) · 2022-1-12

Strengths

1) The manuscript presents a systematic discussion of the three-particle Green's function - i.e. the time ordered product of six fermion operators - for an interacting many-electron system. The calculation is shown in some detail and can be followed relatively easily. 2) The manuscript contains a detailed comparison between various approximations and exact results for a solvable system, the Hubbard dimer. For strongly correlated electron systems that is a very important thing to do in my opinion.

Weaknesses

1) It is somewhat unclear to me inhowfar the method which is presented really is helpful for more complicated systems than a Hubbard dimer. In partiular i wonder if in the case of the inverse photoemission spectrum in the quarter filled ground state - i.e. with one electron in the dimer - taking all states with one added electron and a 'particle hole excitation' of the electron present initially is not equivalent to an exact solution? 2) I somewhat resent the use of the term 'satellites' in this manuscript. For example in the noninteracting three-particle Green's function I would expect that these 'satellites' really are structureless continua and have little to do with the features called satellites in photoemission spectra of correlated electron systems, which are more something like Hubbard bands. 3) I do not understand why the authors are using a Green's function of 6 Fermion operators. Would the most natural extension not be a Green's function that has three fermions at time t_1 and one Fermion at time t_2, i.e. the type of Green's function which shows up in the equation of motion of the single particle Green's function?

Report

Report on
'Photomissionspectroscopy from the three-body Green's function'
by G. Riva et al.

The main point of the manuscript is the discussion of the three-particle
Green's function - i.e. the time ordered product of six fermion operators -
for an interacting many-electron system. The authors derive a spectral
representation which is roughly equivalent to the Lehmann representation
and introduce a special time ordering (Eq. (11) ) to apply this to
photoemssion and inverse photoemission, which is then used to derive the
single-particle Green's function. The authors also introduce a three-particle
self-energy and a Dyson-equation by which the interacting three-particle
Green's function is expressed in terms of the one for noninteracting
particles. As a benchmark the authors then compute various Green's functions
in different approximations for the exactly solvable system of a Hubbard dimer
and compare the results.
The manuscript presents a remarkable amount of work but is somewhat hard to
read. It is not really clear to me inhowfar the method which is presented
really is helpful for more complicated systems than a Hubbard dimer.
Still, I think the manuscript meeets the acceptance criteria once a few minor
corrections have been made as detailed below.

Requested changes

1) Three-particle Green's functions are being studied for a very long time. For example the well-known Hubbard-operators are nothing but products of three Fermion operators. More generally, composite operators have been used for a long time, see PHYSICAL REVIEW B104, 155128 (2021) for a recent example. It would appear to me that these works are more physically motivated than the rather technical approach of the authors and in any way should be mentioned.

2) The derivation of the spectral representation is rather unpleasant to read because some equations are only in section 2.1., others only in Appendix A, so that a lot of back-and-forth scrolling is necessary if one wants to follow the calculation. I would suggest to change this.

The authors should comment on the three points in the 'weak points' section

---

## Round 2 · Author Response

Dear Editor,

Thank you for sending us the reports of the referees.

We thank the referees for their careful reading of the manuscript and for their questions and comments.
We are pleased to read that the referees consider the manuscript as "well written and potentially of strong interest for the community" and "an important addition to the literature".

Both referees ask for some points to be clarified.
In the following we do so and we provide a list of changes.
We also address the comments of the editor-in-charge.

We hope that our revised manuscript will be suitable for publication in SciPost Physics.

Sincerely, the authors.

REVIEWER 1:

Reviewer 1 considers our manuscript as ``well written and potentially of strong interest for the community".
Reviewer 1 would like us to address the following two points.

Reviewer's comment:

"a satellite in photoemission arise from the interaction between electrons. In the non-interacting limit, only single-particle poles exist. It is ok that the G3 contains extra poles also in the non-interacting case, but such poles should not contribute to the ARPES spectral function, i.e. they should have zero intensity. Only interaction, i.e. a static self-energy, could give finite intensity to these poles. The authors find these poles in the analytical description (eq. 24), and they seem to suggest that they contribute to ARPES also in the non-interacting limit. Indeed there is a pole, which they call satellite, in Fig. 1. This point should be clarified."

Our reply:

We agree with the referee that in the ARPES spectral function the satellite amplitudes are non-zero only when the interaction is switched on. This spectral function is defined as the imaginary part of the one-body Green's function (1-GF), for which satellite amplitudes are zero when the interaction is switched off. The confusion stems from the fact that in figures 1-3 we had also reported the imaginary part of the three-body Green's function (3-GF), i.e., without the contraction to get the 1-GF. The imaginary part of the 3-GF is not equal to the ARPES spectral function. The 3-GF contains more information than the 1-GF. Therefore the imaginary part of the 3-GF has non-vanishing satellite amplitudes also in the non-interacting case. The 1-GF are obtained using the contractions in equations (34) and (35) and from the 1-GF the ARPES spectral function can be obtained. Nevertheless, for analysis purposes, it is convenient to introduce a 3-body spectral function as the imaginary part of the 3-GF.
To clarify these points in the revised manuscript we have modified section II.D by adding the following two paragraphs:

Since the spectral representation of $G_3^{e+h}(\omega)$ given in Eq. (12) is similar to the one of $G_1$
it is convenient to introduce a 3-body spectral function for $G_3^{e+h}(\omega)$ that is similar to the spectral function corresponding to $G_1$.
The latter is defined as
\begin{equation}
A(x_1,x_{1'};\omega)=\frac{1}{\pi}\text{sign}(\mu-\omega)\text{Im} G_1(x_1,x_{1'};\omega).
\end{equation}
We can thus define the spectral function $A_3(\omega)$ corresponding to $G_3^{e+h}(\omega)$ according to
\begin{equation}
A_3(\omega)=\frac{1}{\pi}\text{sign}(\mu-\omega)\text{Im} G_3^{e+h}(\omega),
\end{equation}
where, for notational convenience, the spin-position arguments are omitted.

and

We note that the 3-body spectral function is not the spectral function that corresponds to photoemission spectroscopy. Both spectral functions have the same poles but the corresponding amplitudes are different.
In particular, in the non-interacting case the amplitudes of satellites can be non-zero in the 3-body spectral function.
To retrieve the spectral function that corresponds to photoemission spectra Eq. (17) has to be used.

Moreover, figures 1-3 have been modified to emphasize more clearly the differences between the 1- and 3-body spectral functions. These figures are now divided into two panels; in the upper panel we report the 1-body spectral function (the ARPES spectral function), while the 3-body spectral function is reported in the bottom panel.
We included this latter case to show that the 3-GF has the poles at the same position as the 1-GF and that only the amplitudes differ.

Reviewer's comment:

"An important approximation of standard approaches based on G1 (especially in the ab inito community) is to take the diagonal only component of the self-energy, i.e. only the poles are corrected with respect to the zero-order (usually DFT) simulation. G3 has six indexes, and it is not easy to understand where such approximation would enter. This is also related to how demanding would be the inversion of G3 in the QP basis set (appendix C), and which kind of satellites the approach could give. I would expect that, the contraction of the external indexes (indexes im in eqs. 33-34) could give the QP approximation. However, for correlated satellites (plasmons, excitons, magnons, etc .. ) the internal indexes should be allowed to mix. Side comment, something is wrong in the indexes of eq. 30. The Hubbard dimer does not help much here. At least a general discussion in this direction would be useful."

Our reply:

We thank the referee for raising this very interesting point. Indeed the diagonal approximation to the self-energy is an important practical tool to calculate the poles of the one-body Green's function. If the self-energy is dynamical both quasi-particle energies and satellites can be obtained with this approximation, although in practice mainly quasi-particle energies are calculated. Instead, if the self-energy is static only quasi-particles can be calculated.
A diagonal approximation can also be made for the three-body self-energy and this could be very interesting because it would reduce the numerical cost of the calculations significantly. Moreover, from a static three-body self-energy both quasi-particles and satellites can be obtained. Of course, the quality of the quasi-particle energies and satellites not only depend on the diagonal approximation but also on the approximation to the self-energy. The latter approximation is probably the most crucial. As alluded to by the referee we can not test the diagonal approximation on the Hubbard dimer since the three-body self-energy is already diagonal in the basis that diagonalizes the non-interacting 3-GF. The referee mentions the contraction of the external indices in eqs. 33-34 (eqs. 34-35 in the revised manuscript), i.e., $m=i$. However, the equivalent of the diagonal approximation in the three-body case would be to set $m=i$, $o=j$ and $k=l$.

We have now clarified this point in the revised manuscript by adding the following sentences to the outlook given in section 4.

Moreover, we can reduce the numerical cost of the calculations by applying a diagonal approximation to the three-body self-energy, in similar manner as is often done for the one-body self-energy, to calculate only the poles of $G_3^{e+h}$. While a static one-body self-energy can only yield poles that correspond to quasi-particles, a diagonal static three-body self-energy would yield the poles corresponding to both quasi-particles and satellites.

We have also corrected the indices in Eq. (30) (Eq. (31) of the revised manuscript).

REVIEWER 2:

Referee 2 considers our manuscript ``an important addition to the literature" and referee 2 recommends publication after two points are clarified.

Reviewer's comment:

"In Figs. 1-5 can it be clarified whether a curve labeled G3, or Sigma3, is the 3-particle spectral function or the 1-particle spectral function obtained from a calculation of G3? Hopefully the latter."

Our reply:

We thank the referee for this question. This point was indeed not clear.
In the revised manuscript we have modified figures 1-3 to emphasize more clearly the differences between the 1- and 3-body spectral functions. These figures are now divided into two panels; in the upper panel we report the 1-body spectral function (the one that corresponds to photoemission spectra), while the 3-body spectral function is reported in the bottom panel. We included this latter case to show that the 3-GF has the poles at the same position as the 1-GF and that only the amplitudes differ. In figures 4 and 5 only spectral functions corresponding to the 1-GF are shown.

Reviewer's comment:

"Can the actual steps in the calculation of the spectral function using their simplified G3 (static Sigma3) be set out in the language of a general many-electron system and of many-body perturbation theory? E.g. as a flowchart. This would help the reader assess how realistic the Hubbard model is here as a prototype many-electron system. Is Sigma3 spatially non-local? What is the extent of self-consistency imposed by the calculation? To what extent can the recommended procedure be regarded as a true perturbation theory, and if it can, what is the small quantity?"

Our reply:

The exact 3-body self-energy is defined by the Dyson equation given in Eq. (18) and it is indeed a non-local quantity. We do not yet have an expression for the 3-body self-energy in which it is given as an explicit functional of $G_3$ and the interaction. Therefore, in this work we are not doing many-body perturbation theory. For our application to the Hubbard dimer we have derived the exact $G_3^{e+h}$ and $G_{03}^{e+h}$. Therefore, we know the exact $\Sigma_3$ because we can solve the Dyson equation (Eq. (18)) and we can take the static approximation by setting $\omega=0$. To obtain the corresponding $G_3^{e+h}$ we solve once more the Dyson equation in Eq. (18). As a consequence self-consistency is not an issue here. The main goal of this manuscript is to show that, with a static self-energy, the 3-GF has information about satellites, contrary to the 1-GF. We are now trying to make the theory applicable to real systems, by looking for approximations to $\Sigma_3$ as explicit functionals of $G_3^{e+h}$ and the interaction. This interaction, which could be, for example, the bare Coulomb interaction or the screened Coulomb interaction, will then be the small parameter. This is, however, beyond the scope of this work.
We have now clarified these points in the conclusions by adding the following paragraph,

For the specific case of the Hubbard dimer we were able to obtain the exact $G_3^{e+h}$. Therefore we could obtain an exact three-body self-energy by solving a Dyson equation. However, in general, the exact three-body self-energy is unknown. Therefore, our next goal is to derive a general static approximation for the three-body self-energy. This could be achieved, for example, by using the equation of motion for $G^{e+h}_3$ along the same lines as has been done for $G_1$ or by using a similar strategy as in Ref. [27], where a practical scheme to calculate $G_3$ for the description of Auger spectra is proposed.

We also added the following sentences to the outlook given in section 4 briefly discussing a possible strategy to reduce the numerical cost of the calculations, namely by using a diagonal approximation to the three-body self-energy, as was mentioned by Reviewer 1.

Moreover, we can reduce the numerical cost of the calculations by applying a diagonal approximation to the three-body self-energy, in similar manner as is often done for the one-body self-energy, to calculate only the poles of $G_3^{e+h}$. While a static one-body self-energy can only yield poles that correspond to quasi-particles, a diagonal static three-body self-energy would yield the poles corresponding to both quasi-particles and satellites.

EDITOR-IN-CHARGE:

We now address the comments and questions of the editor-in-charge.

Editor's comment:

``The solution of the three-body green's function was already used in the literature: for the Auger problem by A. Marini and M. Cini in Journal of Electron Spectroscopy and Related Phenomena 127 (2002) 17–28 and by C. Calandra and F. Manghi in Phys. Rev. B 50, 2061 to study satellite structures and the occurrence of the metal-insulator transition. I think the authors should mention these two works in their manuscript".

Our reply:

We thank the editor for these two important references, which we have now added to the manuscript in the Introduction, as:

We note that the three-body Green's function has been employed to describe Auger spectra [27] and to study satellite structures and the occurrence of the metal-insulator transition. [28]

Editor's comment:

"Another physical phenomenon that could be studied with the present approach is probably "trions". May the author comment on this possibility? Do they expect trions will be well described by solving the G3 problem?"

Our reply:

We agree with the editor that trions would be another interesting applications of our approach. We had briefly mentioned this possibility in the conclusions. We have now added more references there. We cannot foresee the performance of our approach to describe these excitations because it will depend on the quality on the approximations to the 3-body self-energy. It will be interesting to explore this problem in the future.

Editor's comment:

"I think in Eq. 54 you wrote "$G_2(\omega)$" instead of "$G_1(\omega)$"."

Our reply:

We thank the editor for noticing this typo. We have now corrected it.

---

## Round 2 · List of Changes

We have modified section II.D by adding the following two paragraphs:

Since the spectral representation of $G_3^{e+h}(\omega)$ given in Eq. (12) is similar to the one of $G_1$ it is convenient to introduce a 3-body spectral function for $G_3^{e+h}(\omega)$ that is similar to the spectral function corresponding to $G_1$. The latter is defined as \begin{equation} A(x_1,x_{1'};\omega)=\frac{1}{\pi}\text{sign}(\mu-\omega)\text{Im} G_1(x_1,x_{1'};\omega). \end{equation} We can thus define the spectral function $A_3(\omega)$ corresponding to $G_3^{e+h}(\omega)$ according to \begin{equation} A_3(\omega)=\frac{1}{\pi}\text{sign}(\mu-\omega)\text{Im} G_3^{e+h}(\omega), \end{equation} where, for notational convenience, the spin-position arguments are omitted.

and

We note that the 3-body spectral function is not the spectral function that corresponds to photoemission spectroscopy. Both spectral functions have the same poles but the corresponding amplitudes are different. In particular, in the non-interacting case the amplitudes of satellites can be non-zero in the 3-body spectral function. To retrieve the spectral function that corresponds to photoemission spectra Eq. (17) has to be used.

Moreover, figures 1-3 have been modified to emphasize more clearly the differences between the 1- and 3-body spectral functions. These figures are now divided into two panels; in the upper panel we report the 1-body spectral function (the ARPES spectral function), while the 3-body spectral function is reported in the bottom panel. We included this latter case to show that the 3-GF has the poles at the same position as the 1-GF and that only the amplitudes differ.

2.

We have added the following sentences to the outlook given in section 4.

Moreover, we can reduce the numerical cost of the calculations by applying a diagonal approximation to the three-body self-energy, in similar manner as is often done for the one-body self-energy, to calculate only the poles of $G_3^{e+h}$. While a static one-body self-energy can only yield poles that correspond to quasi-particles, a diagonal static three-body self-energy would yield the poles corresponding to both quasi-particles and satellites.

We have also corrected the indices in Eq. (30) (Eq. (31) of the revised manuscript).

3.

We have added the following paragraph to the conclusions,

For the specific case of the Hubbard dimer we were able to obtain the exact $G_3^{e+h}$. Therefore we could obtain an exact three-body self-energy by solving a Dyson equation. However, in general, the exact three-body self-energy is unknown. Therefore, our next goal is to derive a general static approximation for the three-body self-energy. This could be achieved, for example, by using the equation of motion for $G^{e+h}_3$ along the same lines as has been done for $G_1$ or by using a similar strategy as in Ref. [27], where a practical scheme to calculate $G_3$ for the description of Auger spectra is proposed.

4.

We have added the following sentence to the Introduction,

We note that the three-body Green's function has been employed to describe Auger spectra [27] and to study satellite structures and the occurrence of the metal-insulator transition. [28]

We have added some references of works in the literature that involve the 3-GF

6.

in Eq. 54 we modified "$G_2(\omega)$" to "$G_1(\omega)$"."

---

## Round 3 · Referee Report · Anonymous (Referee 4) · 2022-1-24

Report

With the revisions performed by the authors I think the acceptance criteria now are met.

---

## Round 3 · Referee Report · Anonymous (Referee 3) · 2022-2-1

Report

In their reply the authors have clarified the context of their work and adjusted the manuscript where appropriate. I support publication of the manuscript in SciPost.

---

## Round 3 · Author Response

Dear Editor-in-charge,

Thank you for sending us the reports of the referees.

We thank the referees for their reading of the manuscript and for their questions and comments.
Reviewer 1 writes "I think the manuscript can be now accepted for publication." and Reviewer 2 writes "The authors have made helpful modifications to their manuscript that provide the clarification that I (and, in my opinion, also the other referee and the Editor) requested." Reviewer 4 writes that "the manuscript meets the acceptance criteria once a few minor corrections have been made".
Finally, the comments made by Reviewer 3 seem to indicate that there is a misunderstanding with respect to the objective of our work. We have therefore made several changes to the manuscript including the title which now reads "Photoemission spectral functions from the three-body Green's function." More details are given below.

In the following we address in detail the remaining points raised by the referees.

We hope that our second revision of the manuscript will be suitable for publication in SciPost Physics.

Sincerely, the authors.

REVIEWER 1:

Reviewer's comment:

"The authors have addressed the points raised in my review.
In particular for the definition of the ARPES spectral function from G3. The role of the basis set and the related diagonal approximation remains to be further explored. Future applications on real materials could possibly clarify this.
I think the manuscript can be now accepted for publication."

Authors' response:

We thank the reviewer for recommending our paper for publication in SciPost Physics.

REVIEWER 2:

Reviewer's comment:

"The authors have made helpful modifications to their manuscript that provide the clarification that I (and, in my opinion, also the other referee and the Editor) requested."

Authors' response:

We are glad that the reviewer is happy with the modifications we made to the manuscript.

REVIEWER 3:

Before addressing the comments of the reviewer we would like to make the following general remark.

The reviewer seems to have understood that we want to use the three-body Green's function as fundamental quantity in the calculation of photoemission spectra to include corrections beyond the sudden approximation. This is probably the reason that the reviewer writes that ``these corrections in terms of the three-body Green function needs to be derived in a mathematically consistent fashion from fundamental considerations on the photocurrent."
However this is not the main idea of the paper. We are always working within the sudden approximation and the main goal is to obtain the one-body Green's function, and its corresponding spectral function, from the three-body Green's function. Therefore, to avoid any confusion we have changed the title of our work to "Photoemission spectral functions from the three-body Green's function."

Reviewer's comment:

"The direct and inverse photoemission intensity is given, within the sudden approximation, by the one-electron removal and addition spectra that are proportional to the imaginary part of the retarded single particle Green function (e.g. Rev. Mod. Phys. 75, 473 2003, Phys. Rev. B 94, 115119 2016). This is the case for both (strongly) interacting and non-interacting/weakly interacting systems. This results stems from a direct calculation of the photoelectron current using scattering theory. Beyond the sudden approximation corrections to the photocurrent appear. Hedin and coworkers (Phys. Rev. B 58 15565 1998) have shown that whereas the sudden approximation includes ``intrinsic losses" or satellite structure, adiabatic corrections provide further the ``extrinsic losses"."

Authors' response:

In this work we are always working within the sudden approximation which is the standard in our field. When we write that we use the 3-body Green's function (3-GF) to calculate photoemission spectra it is implied that we mean photoemission spectra within the sudden approximation. It is not mentioned explicitly since it is standard practice in our field. We have now made this point explicit in the second revision of the manuscript in order to avoid any possible misunderstanding by rewriting the following sentence in the introduction:

"The main reason is that the one-body Green's function (1-GF) can be easily linked to photoemission spectra since its poles are the electron removal and addition energies.

as

"The main reason is that the one-body Green's function (1-GF) can be easily linked to photoemission spectra (within the sudden approximation) since its poles are the electron removal and addition energies.

Reviewer's comment:

"The authors state that the three-body Green function contribute to photoemission intensity, but do not derive this statement from a calculation of the photocurrent."

Authors' response:

Nowhere in the manuscript have we made this statement.

Reviewer's comment:

"Instead the existing literature (some of which referred to above) seems to agree that within the sudden approximation (and apart from matrix-element effects), the one-particle Green function contains all spectral information relevant for photoemission. In this context it is not clear how the authors challenge the present status quo and can justify their statement that the three-body Green function is a fundamental quantity to the calculation photoemission spectra. Possibly the implication is that the three-body Green function embodies corrections beyond the sudden approximation. If so, these corrections in terms of the three-body Green function needs to be derived in a mathematically consistent fashion from fundamental considerations on the photocurrent."

Authors' response:

We agree with the reviewer that the one-particle Green function (within the sudden approximation) contains all spectral information relevant for photoemission. However, that is if one has the exact one-body Green's function (1-GF). The main idea of this work is still to calculate the 1-GF, but using the 3-GF to improve the approximations to the 1-GF and, in particular, to capture satellites. This point is explained in detail in the Introduction and Results sections of the paper. In a nutshell, to capture satellites using the standard 1-body approach one requires a dynamical self-energy since the non-interacting 1-GF only contains information about quasi-particles. It is well-known that it is difficult to obtain good dynamical approximations for the self-energy. For example, the $GW$ approximation does not yield very accurate satellites. Instead, when using a 3-body approach, information about satellites is already contained in the non-interacting 3-GF and, therefore, a simpler static 3-body self-energy is sufficient to capture satellites. Once the 3-GF is obtained we contract (according to Eq. (17)) to obtain the 1-GF and therefore the photoemission spectrum including the satellites.
The strategy is similar to that of the Bethe-Salpeter equation. In principle, the one-body polarisation is sufficient to calculate optical spectra but in practice it is more convenient to use a two-body polarisation.
To make this point clearer in the second revision of the manuscript we modified the following sentence in the Introduction:

"Therefore, we will study here the three-body Green's function (3-GF) as the fundamental quantity from which to calculate photoemission spectra.

to

"Therefore, we will study here the three-body Green's function (3-GF) as the fundamental quantity from which to calculate the 1-GF and, hence, photoemission spectra."

REVIEWER 4:

Reviewer's comment:

"It is somewhat unclear to me inhowfar the method which is presented
really is helpful for more complicated systems than a Hubbard dimer.
In partiular i wonder if in the case of the inverse photoemission spectrum in
the quarter filled ground state - i.e. with one electron in the dimer - taking
all states with one added electron and a 'particle hole excitation' of the
electron present initially is not equivalent to an exact solution?"

Authors' response:

This work presents the first step towards an approach based on the 3-GF to calculate accurate spectral functions, namely a proof of principle that with a static 3-body self-energy one can obtain satellites in these spectra.
We show this both by looking at the fundamental equations and by applying our strategy to the Hubbard dimer.
This finding is true for any system, not only for the Hubbard dimer.
As mentioned in the Conclusions, we are currently working on the second step which is to derive a general static approximation for the three-body self-energy.
This could be achieved, for example, by using the equation of motion of the 3-GF.
We agree with the reviewer that only when the second step has finished we will be able to fully assess whether our method is helpful for more complicated systems than a Hubbard dimer.
We think it will almost certainly be the case because, since the 3-GF contains more information than the 1-GF, we can put less information into the 3-body self-energy than in the 1-body self-energy.
This point is indeed nicely illustrated with the Hubbard dimer at 1/4 filling.
Since one cannot have more than three particles in the system, i.e. the added electron and the electron-hole pair which it creates, the exact 3-body self-energy is static since no extra excitations have to be created.
Instead, the exact one-body self-energy is a more complicated dynamical quantity.
We note that our strategy is also similar to that of the Bethe-Salpeter equation. In principle, the one-body polarisation is sufficient to calculate optical spectra but in practice it is more convenient to use a two-body polarisation because it contains more information.

Reviewer's comment:

"I somewhat resent the use of the term 'satellites' in this manuscript.
For example in the noninteracting three-particle Green's function I would
expect that these 'satellites' really are structureless continua and have
little to do with the features called satellites in photoemission spectra
of correlated electron systems, which are more something like
Hubbard bands."

Authors' response:

The poles of the 3-GF (its electron-electron-hole and hole-hole-electron parts, to be precise) are the same as those of the 1-GF but the amplitudes corresponding to these poles are not.
The amplitudes of both quasi-particles and satellites found in photoemission spectra are then obtained by using the contractions in Eq. (17) to obtain the 1-GF from the 3-GF, and thus the one-body spectral function.
Therefore, although the 3-GF does not directly yield information about the amplitudes of the satellites (one has to do the contraction first) it contains all the necessary information about the satellites.
For this reason we have also used the term "satellite" when discussing the 3-GF.
We have made several modifications in the second revision of the manuscript to be more careful when using the term "satellite".
In particular we now write that the noninteracting 3-GF contains information about satellites.

Reviewer's comment:

"I do not understand why the authors are using a Green's function of
6 Fermion operators. Would the most natural extension not be a Green's function
that has three fermions at time $t_1$ and one Fermion at time $t_2$,
i.e. the type of Green's function which shows up in the equation
of motion of the single particle Green's function?"

Authors' response:

The referee is right that using a 2-body Green's function of the form

$\langle\Psi_0^N| (\psi^{\dagger}\psi\psi)_{t_1}|\Psi_n^{N+1}\rangle\langle\Psi_n^{N+1}|(\psi^{\dagger})_{t_2}|\Psi_0^N\rangle$ we have indeed information about addition energies (or removal energies if we consider another order of the field operators). However the non-interacting two-body Green's function (which is the product of two non-interacting 1-GF) corresponding to these times would only have information about quasi-particles, but not about satellites.
Therefore, a static 2-body self-energy would yield a photoemission spectrum without satellites.

Reviewer's comment

"Three-particle Green's functions are being studied for a very long time.
For example the well-known Hubbard-operators are nothing but products of
three Fermion operators. More generally, composite operators have been used
for a long time, see PHYSICAL REVIEW B104, 155128 (2021) for a recent example.
It would appear to me that these works are more physically motivated than
the rather technical approach of the authors and in any way should be mentioned."

Authors' response

We have cited the work mentioned by the reviewer as well as another recent work (Physics Reports 929, 1 (2021))

Reviewer's comment

"The derivation of the spectral representation is rather unpleasant to
read because some equations are only in section 2.1., others only in Appendix A,
so that a lot of back-and-forth scrolling is necessary if one wants to follow
the calculation. I would suggest to change this."

Authors' response

We have added all the relevant equations to Appendix A to avoid the back-and-forth scrolling.

---

## Round 3 · List of Changes

1) We changed the title to "Photoemission spectral functions from the three-body Green's function"

2) We rewrote the following sentence in the introduction as

"The main reason is that the one-body Green's function (1-GF) can be easily linked to photoemission spectra (within the sudden approximation) since its poles are the electron removal and addition energies.

3) We rewrote the following sentence in the introduction as

"Therefore, we will study here the three-body Green's function (3-GF) as the fundamental quantity from which to calculate the 1-GF and, hence, photoemission spectra."

4) We have made several modifications in the second revision of the manuscript to be more careful when using the term "satellite". In particular we now write that the noninteracting 3-GF contains information about satellites.

5) We have cited the work mentioned by Reviewer 4 (PHYSICAL REVIEW B104, 155128 (2021)) as well as another recent work (Physics Reports 929, 1 (2021))

6) We have added all the relevant equations to Appendix A to avoid the back-and-forth scrolling.

7) We corrected eqs.(21-24) that presented an additional imaginary number in the denominator

8) We added the complex conjugate symbols that were missing in eq.(31).

---

## Editorial Decision

published